# FedFFT: Taming Client Drift in Federated SAM via Spectral Perturbation Filtering

## Abstract

Federated Learning (FL) enables decentralized training without data sharing, but suffers from statistical heterogeneity across clients, leading to client drift, poor generalization, and sharp minima compared to centralized training. Sharpness-Aware Minimization (SAM) has emerged as a promising approach to improve generalization, yet its application in federated learning still suffers from divergence problems, since perturbations are computed locally and reflect client-specific loss geometries. To better understand this issue, we provide experimental evidence from a new perspective—the frequency domain—for SAM perturbations in federated settings, revealing that inter-client perturbation inconsistencies are predominantly concentrated in the low-frequency spectrum. Motivated by this insight, we propose **Fed**erated learning with **F**requency-domain **F**iltering of SAM per**t**urbations (**FedFFT**). It is a lightweight and plug-and-play method that filters out low-frequency components of SAM perturbations without requiring additional communication, thereby suppressing inconsistent components in client updates while preserving consistent learning signals. Extensive experiments across multiple benchmarks and diverse backbones demonstrate that FedFFT consistently outperforms SAM-based FL methods, particularly under severe non-IID distributions. These results highlight the effectiveness, scalability, and general applicability of our frequency-domain perspective for sharpness-aware federated optimization.

## 1 Introduction

Federated Learning (FL) (McMahan et al., 2017) is a distributed learning paradigm where multiple clients collaboratively train a global model under the coordination of a central server, while keeping their raw data local to preserve privacy. In each communication round, clients perform local training and only transmit model parameters or updates, which are aggregated to update the global model. This framework has been widely applied in privacy-sensitive domains such as healthcare, finance, mobile applications, and autonomous systems (Antunes et al., 2022; Rauniyar et al., 2024; Fantauzzo et al., 2022). However, the practical effectiveness of FL is severely hampered by the statistical heterogeneity inherent in real-world data, where the local data distributions across clients are typically non-independent and identically distributed (Non-IID). This causes the optimization objectives of individual clients to become misaligned with one another and with the global goal, resulting in local updates that pull the shared model in conflicting directions. This phenomenon are known as client drift (Karimireddy et al., 2021; Woodworth et al., 2020; Li et al., 2020a; Fan et al., 2022), which not only slows convergence but can also limits the model's ability to generalize to the overall underlying distribution.

To address the generalization challenges posed by client drift, a promising research direction has shifted from traditional client-side regularization (Acar et al., 2021; Karimireddy et al., 2021; Li et al., 2020b; 2021b; Xu et al., 2021a) or aggregation methods (Ye et al., 2023; Li et al., 2023; Shi et al., 2025) to exploring the geometry of the loss landscape. These methods build upon the insight that convergence to sharp minima correlates with poor generalization (Li et al., 2020a), where flatter minima often yield better performance. Sharpness-Aware Minimization (SAM) (Foret et al., 2020) is a representative technique designed for this purpose, seeking flatter regions by optimizing a perturbed loss. Building on this, FedSAM (Qu et al., 2022; Caldarola et al., 2022) pioneered the application of SAM to local training in Federated Learning. While FedSAM (Sun et al., 2023a;

Fan et al., 2024) have shown strong performance across different settings, they primarily focus on local flatness, implicitly assuming that minimizing sharpness locally leads to a globally flat minimum. In practice, however, under substantial data heterogeneity, the local and global loss landscapes may diverge considerably, and improvements in local flatness do not necessarily guarantee global flatness. Subsequent works (Sun et al., 2023a; Fan et al., 2024; Caldarola et al., 2025; Dai et al., 2024) have attempted to bridge this local-global gap through various strategies, such as enhancing client-side updates (Sun et al., 2023a) or shifting the sharpness optimization to the server (Caldarola et al., 2025), sometimes employing complex frameworks like the Alternating Direction Method of Multipliers (ADMM) (Boyd et al., 2011). Despite these advances, existing solutions face a difficult trade-off: purely client-side methods struggle with the deviation of local sharpness estimates, while server-centric approaches often come at the cost of significant communication or computational overhead. Crucially, none of these approaches explicitly investigate the intrinsic nature of the perturbations themselves, leaving open the question of whether they can be refined to better align clients with the global objective.

In this work, we address this question by introducing a novel frequency-domain perspective. To our knowledge, we are the first to conduct a systematic experimental study on the spectral properties of client-side SAM perturbations in FL. Our key finding is that the inter-client disagreements are not random noise; they are predominantly concentrated in the low-frequency spectrum, which we hypothesize is strongly tied to client-specific data biases. Motivated by this insight, we propose Federated learning with Frequency-domain Filtering of SAM perturbations (FedFFT), a lightweight and plug-and-play method. FedFFT applies a high-pass filter to the locally computed perturbations, systematically removing the discordant low-frequency components while preserving consistent learning signals in the higher frequencies. This alignment in the spectral domain is achieved without requiring any additional communication overhead. Notably, FedFFT can be seamlessly integrated as a plug-and-play module with federated learning framework that utilizes client-side SAM optimizers, further broadening its applicability. Our contributions are summarized as below: **(1) Frequency-domain analysis of perturbations.** We provide the first study of SAM perturbations in FL across clients based on spectral decomposition, and reveal that heterogeneity is primarily concentrated in the low-frequency bands. **(2) Algorithm design.** Based on this insight, we introduce FedFFT, a simple yet effective approach that filters out low-frequency perturbation components to suppress inconsistent client updates while retaining retaining consistent learning signals. **(3) Extensive empirical validation.** We conduct comprehensive experiments across multiple benchmarks and backbones, under varying degrees of data heterogeneity. The results show that FedFFT outperforms related baselines, particularly in highly non-IID settings, demonstrating both effectiveness and scalability.

## 2 RELATED WORK

**Sharpness-Aware Minimization (SAM).** The connection between generalization and flat minima was first recognized in early studies (Hochreiter & Schmidhuber, 1994), and later work confirmed that smoother loss landscapes generally lead to better generalization (Keskar et al., 2017; Neyshabur et al., 2017). Building on this insight, Sharpness-Aware Minimization (SAM) was introduced as a PAC-Bayesian inspired method that explicitly minimizes loss sharpness and achieves strong generalization across image classification benchmarks (Foret et al., 2020). Since then, numerous extensions have been developed. A scale-invariant version improves training stability (Kwon et al., 2021), while another reformulates sharpness from both theoretical and intuitive perspectives (Zhuang et al., 2022). Further studies focus on perturbation strategies, including adaptive or random amplitudes (Liu et al., 2022; Ahn et al., 2024), dynamic adjustment through DSAM (Chen et al., 2024), and variance reduction across domains with DISAM (Zhang et al., 2024).

**Federated Learning.** Federated Learning (FL), introduced with FedAvg (McMahan et al., 2017), enables collaborative model training without raw data sharing. While preserving privacy, this decentralized design exacerbates the *client-drift problem*—the divergence between local and global updates—mainly due to non-IID data and multi-step local errors (Acar et al., 2021; Woodworth et al., 2020; Li et al., 2020a). Limited client participation further aggravates drift and degrades performance. To mitigate client drift, existing methods can be broadly grouped into two categories: (i) *local objective regularization*, which modifies local training to align client updates with the global objective, such as SCAFFOLD (Karimireddy et al., 2021) , FedProx (Li et al., 2020b) and FedDyn

(Acar et al., 2021); and (ii) *modified aggregation strategies*, which design more robust global update rules beyond simple averaging, such as FedAWA (Shi et al., 2025), FedLAW (Li et al., 2023), and FedDisco (Ye et al., 2023). While these approaches improve optimization stability, they are primarily rooted in empirical risk minimization and often overlook the relationship between the global loss landscape and generalization ability. This motivates a new research line that leverages SAM in FL.

**SAM in Federated Learning.** FedSAM (Qu et al., 2022; Caldarola et al., 2022) first brought SAM into FL by applying local perturbations to improve generalization. Subsequent variants extended this idea. For example, FedSpeed (Sun et al., 2023b) used an Alternating Direction Method of Multipliers (ADMM) framework to enhance communication efficiency. PLGU (Qu et al., 2023) and FedSOL (Lee et al., 2024), explored layer-wise perturbation and proximal-based corrections. FedGAMMA (Dai et al., 2024) introduced variance-reduction techniques to align client updates from a global perspective. FedSMOO (Sun et al., 2023a) reduced inconsistency by correcting both updates and perturbations through ADMM, while FedLESAM (Fan et al., 2024) introduced global perturbations to better guide local training. FedFSA (Xing et al., 2025) focused on parameter sensitivity, applying stronger perturbations only to the most sensitive layers to balance convergence and generalization. FedGloSS (Caldarola et al., 2025) shifted attention from local sharpness to global flatness by applying SAM on the server, highlighting the importance of global geometry in federated optimization. Unlike these approaches, our method is motivated by a novel spectral analysis, revealing inter-client disagreement in low-frequency components, which we filter to produce consistent perturbations and flatter minima. This also differs from Fourier-based domain generalization approaches such as FBF (Xu et al., 2021b), which apply frequency transforms to input images; in contrast, we operate in parameter space in FL and modify optimizer dynamics rather than data distributions.

## 3 BACKGROUND

### 3.1 SHARPNESS-AWARE MINIMIZATION

To improve model generalization and robustness, modern optimization methods have shifted focus from merely finding solutions with low training loss to finding solutions that reside in flat minima of the loss landscape. SAM (Foret et al., 2020) is a leading technique for this purpose. It jointly minimizes the loss value and the sharpness by solving the following minimax objective:

$$\min_{w} \max_{\|\delta\| \leq \rho} \mathcal{L}(w + \delta), \tag{1}$$

where $\mathcal{L}(\cdot)$ is the empirical loss on the training data, and $\rho$ is the neighborhood size. In practice, the inner maximization is approximated with a single step of gradient ascent. The full optimization process for parameter $w$ involves two steps: (1) Compute the perturbation that approximately maximizes the loss: $\delta^*(w) = \rho \frac{\nabla \mathcal{L}(w)}{\|\nabla \mathcal{L}(w)\|_2}$; (2) Update the model parameters using the gradient at the perturbed point: $w \leftarrow w - \eta \nabla \mathcal{L}(w + \delta^*(w))$. This procedure encourages the optimizer to converge to flat minima, which are empirically linked to better generalization performance.

### 3.2 FEDERATED LEARNING

FL is a distributed learning paradigm that enables training a global model on data from $K$ clients, coordinated by a central server, without centralizing the private client datasets $\mathcal{D}_k$. The core objective in FL is to minimize the global empirical risk $F(w)$, defined as the weighted average of the local empirical losses $f_k(w)$:

$$\min_{w} F(w) := \frac{1}{K} \sum_{k=1}^{K} f_k(w), \quad \text{where} \quad f_k(w) = \frac{1}{|\mathcal{D}_k|} \sum_{(x,y) \in \mathcal{D}_k} \mathcal{L}_k(w; x, y). \tag{2}$$

The widely-adopted FedAvg algorithm (McMahan et al., 2017) solves this objective via iterative communication rounds. In each round, the server **(1) Broadcasts** the global model $w^t$ to clients. Clients then perform **(2) Local Updates** on their data to produce $w_k^{t+1}$. These are **(3) Uploaded** to the server for **(4) Aggregation** into the new global model $w^{t+1} = \sum_k p_k w_k^{t+1}$.

A key challenge in FL is **data heterogeneity**, where client data distributions are Non-Independent and Identically Distributed (Non-IID). This causes the local objectives $f_k(w)$ to be inconsistent with

one another, leading to misaligned loss landscapes and the "client drift" phenomenon, which poses a significant challenge to training a robust global model.

### 3.3 SAM IN FEDERATED LEARNING

Given the challenge of training on misaligned local landscapes, a natural strategy is to seek solutions in flat minima. This motivates applying SAM to FL, known as FedSAM (Qu et al., 2022), which incorporates SAM into each client's local training. The local and global objectives are:

$$\min_{w} F^{\text{SAM}}(w) := \frac{1}{K} \sum_{k=1}^{K} f_k^{\text{SAM}}(w), \quad \text{where} \quad f_k^{\text{SAM}}(w) = \max_{\|\delta_k\| \le \rho} f_k(w + \delta_k). \tag{3}$$

Compared with FedAvg, FedSAM differs in that it optimizes the sharpness-aware local objectives rather than the original local losses. While FedSAM encourages convergence to locally flat regions, it does not guarantee a flat global landscape. Under data heterogeneity, the locally computed perturbation vectors $\delta_k$, which capture the directions of sharpness, can themselves diverge. This raises a critical question that existing works have not fully explored: **what is the underlying structure of these inter-client perturbation disagreements?** Understanding this is key to mitigating their negative impact, which directly motivates our work.

## 4 OUR PROPOSED METHOD

In this section, we introduce our proposed method, Federated learning with Frequency domain Filtering of SAM perturbations (FedFFT). We begin by elaborating on the motivation stemming from our observations of client perturbations in federated environments. Subsequently, we provide a detailed description of the FedFFT algorithm.

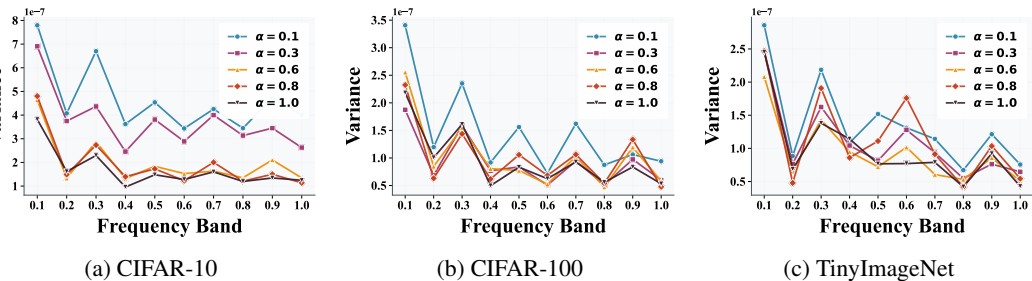

| (a) CIFAR-10 | (b) CIFAR-100 | (c) TinyImageNet |

Figure 1: (a–c) Variance of SAM perturbations across clients under different levels of data heterogeneity, controlled by the Dirichlet concentration parameter $\alpha$. Across all datasets and settings, inter-client variance is consistently concentrated in the low-frequency components, indicating that client disagreement is primarily a low-frequency phenomenon. For a detailed explanation of the Dirichlet partitioning scheme and the role of $\alpha$, please refer to Appendix C.5.

### 4.1 MOTIVATION: ANALYZING CLIENT PERTURBATIONS IN THE FREQUENCY DOMAIN

In FL, non-IID data distributions among clients are the primary cause of the client drift phenomenon. While Sharpness-Aware Minimization (SAM) is a powerful technique for improving generalization, its application in FL may amplify this issue. Since SAM perturbations are computed locally, they intrinsically reflect the geometry of client-specific loss landscapes, causing the perturbation vectors themselves to diverge. However, the underlying structure of these perturbation divergences remains poorly understood. To bridge this gap, we introduce a novel diagnostic approach.

To the best of our knowledge, this work is the first to employ frequency-domain analysis to systematically investigate the nature of SAM perturbation disagreements in a federated context. We treat each client's perturbation vector as a signal and use the Real-valued Fast Fourier Transform (RFFT) to observe its characteristics. Our analysis is conducted on a per-layer basis to respect the model's architectural integrity, with the full implementation details provided in below Section 4.2. For each

layer, we compute the variance across clients within different frequency bands. As illustrated in Figure 1, which averages these variances across all layers, we discover a clear pattern:

*The primary disagreements are concentrated in the low-frequency components, while the disagreements in high-frequency components are much smaller. This finding suggests that the low-frequency parts are highly correlated with client-specific biases, whereas the high-frequency parts may represent more consistent features of the shared learning task.*

Based on this key insight, we formulate our central hypothesis: by filtering out the discordant low-frequency components while preserving the consistent high-frequency information, we can mitigate client drift and improve the global model's performance.

## 4.2 FedFFT: The Proposed Method

Considering that inter-client disagreements are predominantly a low-frequency phenomenon, we propose **Fed**erated learning with **F**requency-domain **F**iltering of SAM per**t**urbations (**FedFFT**). Our method refines the client-side SAM update by integrating a frequency filtering module. This module is designed as a lightweight, plug-and-play replacement for the standard perturbation calculation within any federated learning framework that employs SAM-based optimizers on client devices.

Let's consider a single local update step for a client $k$ at communication round $t \in [1, T]$ and local iteration $e \in [1, E]$. For simplicity, we omit the round and local iteration indices. Now, we first need to compute the standard SAM perturbation $\delta^k$ for client $k$ like FedSAM, denoted as

$$\delta^k = \rho \cdot \frac{\nabla \mathcal{L}_k(w^k)}{\|\nabla \mathcal{L}_k(w^k)\|_2}, \tag{4}$$

where $\delta^k$ has the same size with model parameter $w^k$ and includes all layer-wise perturbations $\delta_{1:L}^k$ and $\delta_l^k$ is the perturbation of parameter $w_l^k$ at layer $l$. The core idea of FedFFT is to **selectively discard the discordant low-frequency components** of the SAM perturbation while preserving the high-frequency components that capture more consistent aspects of the sharpness landscape. This filtering is performed on a per-layer basis to respect the model's architectural integrity. For any given weight layer $l$ with parameters $w_l^k$, the FedFFT procedure is as follows:

1. **Transform to Frequency Domain**: Instead of directly applying this perturbation, we transform it into the frequency domain. We flatten the perturbation tensor $\delta_l^k$ into a vector $\mathbf{v}_l^k$ and apply the Real-valued Fast Fourier Transform (rFFT):

$$\hat{\mathbf{v}}_l^k = \text{rFFT}(\mathbf{v}_l^k), \tag{5}$$

where $\hat{\mathbf{v}}_l^k$ is the frequency-domain representation of the perturbation.

2. **Apply High-Pass Filter**: Next, we apply a high-pass filter operator, $\mathcal{H}_r(\cdot)$, which zeroes out the lowest $r$ fraction of frequency coefficients. Given a truncation ratio $r \in [0, 1)$, the filtering operation is defined as:

$$[\mathcal{H}_r(\hat{\mathbf{v}}_l^k)]_m = \begin{cases} 0, & \text{if } m < \lfloor r \cdot \text{len}(\hat{\mathbf{v}}_l^k) \rfloor, \\ [\hat{\mathbf{v}}_l^k]_m, & \text{otherwise,} \end{cases} \tag{6}$$

where $m$ indexes the frequency coefficients in ascending order. This step is the crux of our method, as it explicitly removes the client-specific biases encoded in the low-frequency domain.

3. **Reconstruct Filtered Perturbation**: We then transform the filtered vector back into the parameter domain using the inverse rFFT (iRFFT) and reshape it to its original tensor shape to obtain the refined perturbation $\tilde{\delta}_l$:

$$\tilde{\delta}_l^k = \text{reshape}(\text{iRFFT}(\mathcal{H}_r(\hat{\mathbf{v}}_l^k))). \tag{7}$$

For simplicity, for any given layer's perturbation tensor $\delta_l^k$ at client $k$, we summarize the three steps above as a filtering operation:

$$\tilde{\delta}_l^k = \text{Filter}(\delta_l^k, r). \tag{8}$$

Finally, the local model in client $k$ is updated using this filtered perturbation, following the standard SAM procedure:

$$w^k \leftarrow w^k - \eta \cdot \nabla \mathcal{L}_k(w^k + \tilde{\delta}^k), \tag{9}$$

where $\tilde{\delta}^k$ is the collection of all layer-wise filtered perturbations $\tilde{\delta}_l^k$.

By replacing the standard SAM perturbation with our filtered version, FedFFT forces the local optimizer to ignore the most heterogeneous directions of sharpness, thereby promoting greater consistency among client updates and facilitating convergence to a more robust global minimum. As shown in, we provide the workflow of FedAvg, FedSAM and our FedFFT in Algorithm 1. Benefit from the effective and general filtering method, ours can also be combined with some classical or advanced FL-based methods; please see more details in Appendix A.

## 5 EXPERIMENTS

### 5.1 EXPERIMENTAL SETUPS

**Datasets and Baselines.** We evaluate on CIFAR-10 (Krizhevsky & Hinton, 2009), CIFAR-100 (Krizhevsky & Hinton, 2009), and Tiny-ImageNet (CS231N, 2015). To simulate data heterogeneity, we partition datasets using a Dirichlet distribution with $\alpha \in 0.1, 0.6$, where smaller $\alpha$ yields more non-IID distributions and larger $\alpha$ yields more IID ones.

We benchmark FedFFT against a comprehensive suite of baselines in three main categories: (i) foundational FL algorithms, including FedAvg (McMahan et al., 2017), SCAFFOLD (Karimireddy et al., 2021), and FedDyn (Acar et al., 2021); (ii) the direct application of SAM in FL, namely FedSAM and its momentum-enhanced variant MoFedSAM (Caldarola et al., 2022; Qu et al., 2022); and (iii) advanced FL-SAM variants that aim to improve consistency, such as FedGAMMA (Dai et al., 2024), FedSMOO (Sun et al., 2023a), FedLESAM (Fan et al., 2024), and Fed-GloSS (Caldarola et al., 2025). To ensure fair comparisons, we integrate our approach with these foundational algorithms following FedLESAM, yielding FedFFT

---

**Algorithm 1** FedAvg, FedSAM and Our FedFFT

**Require:** Communication rounds $T$, local epochs $E$, perturbation radius $\rho$, local learning rate $\eta$, frequency truncation ratio $r$.
**Ensure:** Global model $w_g^T$
1: Initialize global model $w_g^0$.
2: **for** $t = 0$ to $T - 1$ **do**
3:     Randomly select active client set $S_t$.
4:     **for** all clients $k \in S_t$ **in parallel do**
5:         $w^{k,t,0} \leftarrow w_g^t$
6:         **for** $e = 0$ to $E - 1$ **do**
7:             ▷ perturbation stage
8:             FedAvg: $\delta^{k,t,e} = 0$
9:             FedSAM: $\delta^{k,t,e} = \rho \cdot \frac{\nabla \mathcal{L}_k(w^{k,t,e})}{\| \nabla \mathcal{L}_k(w^{k,t,e}) \|_2}$
10:            FedFFT: $\delta_{1:L}^{k,t,e} = \mathrm{Filter}(\delta_{1:L}^{k,t,e}, r)$
11:            $w^{k,t,e+1} = w^{k,t,e} - \eta \mathcal{L}_k(w^{k,t,e} + \delta^{k,t,e})$
12:         **end for**
13:         Send local model $w^{k,t,E}$ to server.
14:     **end for**
15:     $w_g^{t+1} \leftarrow \frac{1}{|S_t|} \sum_{k \in S_t} w^{k,t,E}$
16: **end for**
17: **return** $w_g^T$

---

(FedAvg-based), FedFFT-S (SCAFFOLD-based), and FedFFT-D (FedDyn-based). We summarize the characteristics of SAM in FL methods; please refer to the Appendix B for details.

**Implementation Details.** Following prior works (Sun et al., 2023a; Fan et al., 2024), we adopt ResNet-18 (He et al., 2016) from the PyTorch model zoo (Paszke et al., 2019) as the backbone. We use the following settings: 100 clients with about 10% sampled per round, local/global learning rates of 0.1/1.0, 5 local epochs, and up to 800/800/300 communication rounds for CIFAR-10, CIFAR-100, and TinyImageNet. For SAM-based methods, we use a perturbation radius of 0.1 with SGD, weight decay 1e-3, and exponential LR decay (0.998 per round). For FedFFT, the frequency truncation ratio is set to 0.01. Further details are in Appendix C.

### 5.2 MAIN RESULTS

**Comparison with State-of-the-Art Baselines.** Table 1 presents the results on CIFAR-10, CIFAR-100, and Tiny-ImageNet with ResNet-18. Overall, FedFFT delivers consistent improvements under

Table 1: Test accuracy comparison (%) of different methods on CIFAR-10, CIFAR-100 and Tiny-ImageNet, with ResNet-18. "-S" and "-D" means using SCAFFOLD and FedDyn as the base algorithms. All results on Tiny-ImageNet are reproduced by us. Results marked with † are reproduced by us for CIFAR-10 and CIFAR-100. Others are reported from (Sun et al., 2023a) and (Fan et al., 2024).

| Method | CIFAR-10 | | CIFAR-100 | | Tiny-ImageNet | |
|---|---|---|---|---|---|---|
| | $\alpha = 0.6$ | $\alpha = 0.1$ | $\alpha = 0.6$ | $\alpha = 0.1$ | $\alpha = 0.6$ | $\alpha = 0.1$ |
| FedAvg | 79.52 | 76.00 | 46.35 | 42.64 | 28.31 | 27.48 |
| FedSAM† | 81.91 | 74.92 | 48.08 | 45.53 | 33.16 | 29.46 |
| MoFedSAM | 84.13 | 78.71 | 54.38 | 44.85 | 33.50 | 29.77 |
| FedLESAM | 81.04 | 76.93 | 47.92 | 44.48 | 27.91 | 26.91 |
| Our FedFFT | 83.02 | 77.53 | 48.59 | 46.83 | 33.58 | 30.43 |
| SCAFFOLD | 81.81 | 78.57 | 51.98 | 44.41 | 35.34 | 32.11 |
| FedSAM-S† | 83.88 | 76.68 | 50.19 | 49.14 | 35.84 | 31.73 |
| FedGamma-S | 82.64 | 78.95 | 53.41 | 46.39 | 36.85 | 30.09 |
| FedLESAM-S | 84.94 | 79.52 | **54.61** | 48.07 | 28.47 | 27.70 |
| Our FedFFT-S | 84.69 | 79.24 | 52.75 | 49.85 | 36.15 | 33.08 |
| FedDyn | 83.22 | 78.08 | 50.82 | 42.50 | 28.01 | 24.19 |
| FedSAM-D† | 82.29 | 79.11 | 53.70 | 46.28 | 38.18 | 31.39 |
| FedSMOO-D | 84.55 | 80.82 | 53.92 | 46.48 | 38.71 | 32.45 |
| FedLESAM-D | 84.27 | 80.08 | 53.27 | 46.42 | 27.36 | 25.32 |
| FedGloSS-D† | 82.58 | 79.23 | 50.92 | 47.36 | 31.72 | 28.04 |
| Our FedFFT-D | **87.19** | **83.05** | 54.46 | **50.90** | **40.85** | **34.46** |

both SCAFFOLD and FedDyn frameworks, indicating that frequency-domain perturbation modeling complements existing federated optimization paradigms. On CIFAR-10, FedFFT-D achieves the best performance under both Dirichlet partitions, demonstrating its effectiveness in standard federated settings. For CIFAR-100, while FedFFT-D is slightly outperformed by FedLESAM-S under $\alpha = 0.6$, it regains superiority in the more heterogeneous $\alpha = 0.1$ case. This suggests that suppressing low-frequency perturbations is particularly beneficial when client data distributions are highly skewed. Moreover, on Tiny-ImageNet, FedFFT-D clearly surpasses all baselines, highlighting its robustness and scalability to larger and more challenging tasks. Furthermore, the convergence curves in Figure 6 of Appendix D.1 show that FedFFT not only achieves higher final accuracy but also converges significantly faster. This suggests that by harmonizing client updates at a spectral level, FedFFT facilitates a more direct and stable optimization path toward a high-quality global minimum, reducing wasted communication rounds spent reconciling conflicting updates. Additionally, we have plotted the loss landscapes of different algorithms, which reveal that our method yields a flatter loss landscape. For details, please refer to the Appendix D.2.

**Generalization Across Diverse Model Architectures.** To verify that the efficacy of FedFFT is not confined to a specific model class, we evaluate its performance across a diverse range of architectures, from lightweight ResNet-18, ResNet-20 to deeper DenseNet-121 and Vision Transformers (ViT). We perform on CIFAR-10 and CIFAR-100 with a moderately heterogeneous setting $\alpha = 0.6$. As shown in Tab 2, FedFFT usually outperforms baselines across different architectures. Critically,

Table 2: Test accuracy (%) across different backbones on CIFAR-10 (C10) and CIFAR-100 (C100) with Dirichlet ($\alpha = 0.6$). All methods use FedDyn as the base algorithm.

| Data | Method | ResNet18 | ResNet20 | DenseNet121 | ViT |
|---|---|---|---|---|---|
| C10 | FedSAM | 82.29 | 88.82 | 89.47 | 49.04 |
| | FedSMOO | 84.55 | 89.86 | 88.72 | 50.23 |
| | FedGloSS | 82.58 | 84.17 | 86.84 | 50.10 |
| | FedFFT | **87.19** | **90.56** | **90.57** | **53.31** |
| C100 | FedSAM | 53.70 | 58.92 | **64.19** | 28.01 |
| | FedSMOO | 53.92 | 58.17 | 63.74 | 29.48 |
| | FedGloSS | 50.92 | 46.68 | 57.14 | 27.54 |
| | FedFFT | **54.46** | **61.60** | 61.66 | **30.36** |

the performance gains are substantial even on powerful models like ViT. This suggests that the problem FedFFT addresses—the inconsistency in the low-frequency spectrum of client perturbations—is a fundamental artifact of the federated optimization process itself, independent of a model's representation capacity. Simply using a larger model does not automatically resolve the geometric misalignment between clients. Our spectral filtering acts as a complementary and orthogonal improvement, harmonizing the local updates to allow these powerful architectures to converge more effectively. These results therefore underscore the broad applicability and scalability of FedFFT, establishing it as a model-agnostic enhancement for sharpness-aware federated learning.

## 5.3 FURTHER ANALYSIS

**Ablation on Filtering Strategy.** To validate our central hypothesis—that inter-client inconsistency is primarily concentrated in the low-frequency spectrum—we compare our proposed low-frequency filtering with several alternative perturbation-modification strategies. Beyond high-frequency and random filtering baselines, we further examine two smoothing-style approaches applied to the SAM perturbation

$$\delta_k = \rho \cdot \nabla L_k(w_k)/\|\nabla L_k(w_k)\|_2.$$

*L2-norm rescaling* constrains the perturbation magnitude by projecting $\delta_k$ back to the SAM radius whenever its $\ell_2$-norm exceeds $\rho$. *Coordinate-wise clipping* suppresses extreme perturbation values using a threshold proportional to the median magnitude of $\delta_k$. We evaluate all filtering strategies on CIFAR-10 under Dirichlet heterogeneity with $\alpha = 0.6$ and $\alpha = 0.1$ using a ResNet-18 backbone. As shown in Table 3, high-frequency filtering, random filtering, L2-norm rescaling, and coordinate-wise clipping all produce negligible improvements

Table 3: Ablation of different perturbation filtering strategies on CIFAR-10 (ResNet-18).

| Filtering Strategy | $\alpha = 0.6$ | $\alpha = 0.1$ |
|---|---|---|
| None (FedSAM) | 81.91 | 74.92 |
| High-frequency | 81.66 | 74.81 |
| Random filtering | 81.97 | 75.09 |
| L2-norm rescaling | 81.88 | 74.79 |
| Coordinate-wise clipping | 80.13 | 75.19 |
| Low-frequency (Ours) | **83.02** | **77.53** |

over the standard FedSAM baseline. In contrast, removing low-frequency components consistently yields substantial and stable performance gains across both heterogeneity levels. These results demonstrate that the effectiveness of FedFFT does not stem from generic smoothing, coefficient truncation, or norm-based regularization, but specifically from filtering the low-frequency spectral components of SAM perturbations—the components exhibiting the largest cross-client disagreement under heterogeneous federated settings.

**Impact of Filtering Ratio.** We explore the impact of varying the filtering ratio $r$ from $0.1\%$ to $8\%$ to highlight the flexibility and adaptability of hyperparameter tuning in our proposed FedFFT. As shown in Fig. 2, we report the performance of different $r$ values in FedFFT, where we view FedSAM as baseline and use FedAvg, SCAFFOLD and FedDyn as the base algorithms, respectively. We can find that even with a small $r$ such as $0.1\%$, ours can consistently outperform its baseline. It proves that removing the discordant low-frequency components is beneficial for sharpness-aware federated optimization. Besides, high filtering ratios ($> 7\%$) lead to gradual performance degradation, approaching or slightly falling below the baseline, which may remove useful information for optimization. Overall, these re-

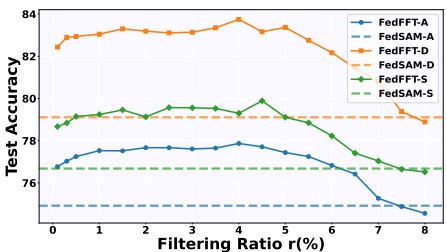

Figure 2: Test accuracy with different filtering ratios on CIFAR-10 ($\alpha = 0.1$). "-A", "-S" and "-D" mean using FedAvg, SCAFFOLD and FedDyn as the base algorithms.

sults confirm that our FedFFT can stably improve accuracy across all models when maintaining a reasonable filtering ratio. Based on our experiments, we recommend a safe range for $r$ between $0.5\%$ and $4\%$.

**Robustness to FL Settings.** We further assess the robustness of FedFFT across three critical federated learning hyperparameters: client activation rate, number of local epochs, and total number of clients. As summarized in Fig. 3, FedFFT consistently and significantly outperforms FedSAM across all tested configurations. Regarding client participation in Fig. 3 (a), FedFFT maintains a stable performance advantage even with a low activation rate of 5%, a challenging scenario that often exacerbates client drift. Similarly, when varying the number of local epochs in Fig. 3 (b), FedFFT demonstrates larger gains with fewer epochs, suggesting faster convergence, while still maintaining a clear edge with more local training. Finally, Fig. 3 (c) shows that our method's superiority holds as the number of clients scales, confirming its applicability to large-scale federated networks. These results collectively demonstrate that the benefits of spectral filtering are not confined to a specific setting but are robust to the practical constraints of real-world FL systems.

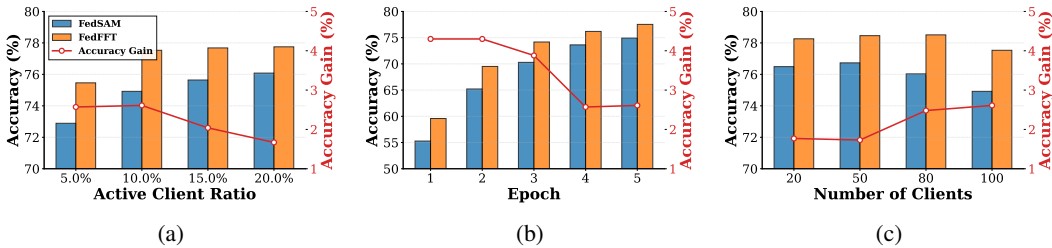

(a)   (b)   (c)

Figure 3: Comprehensive performance comparison between FedFFT and FedSAM across diverse experimental configurations on CIFAR-10 (Non-IID $\alpha = 0.1$) using ResNet-18 backbone.

**Visualizing the Effect of Spectral Filtering.**   To qualitatively understand how FedFFT achieves greater consistency, we visualize the distributions of client perturbations, model features, and model parameters. The visualizations in Fig. 4 reveal a clear causal chain. *(1) More Cohesive Perturbations (Fig. 4a):* The process begins with the perturbations themselves. The original SAM perturbations from different clients are widely scattered in the PCA space. After applying FedFFT's low-frequency filter, these perturbations become significantly more compact, confirming that our method successfully reduces inter-client discrepancy at its source. *(2) Aligned Feature Representations (Fig. 4b):* This improved consistency in perturbations directly translates to more aligned model behavior. We observe that the average features extracted by the FedFFT-trained model are much more tightly clustered for each class compared to the scattered features from the FedSAM model. *(3) Consolidated Client Models (Fig. 4c):* Ultimately, this leads to better convergence of the models themselves. The parameters of client models trained with FedFFT exhibit a much smaller variance and are clustered closer to a central point, indicating that spectral filtering effectively mitigates client drift and guides all clients toward a more unified and robust global solution.

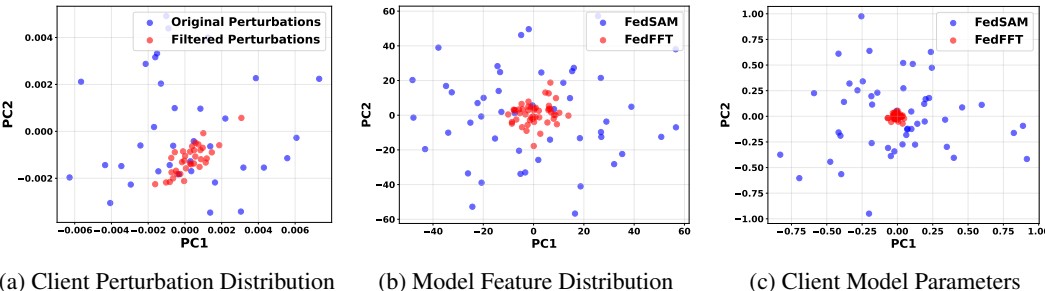

(a) Client Perturbation Distribution   (b) Model Feature Distribution   (c) Client Model Parameters

Figure 4: Effect of low-frequency perturbation filtering on ResNet-18/CIFAR-10 ($\alpha = 0.1$). The comparison between FedSAM (blue) and FedFFT (red) shows: (a) increased compactness of client perturbations (`conv1.weight`), leading to (b) more consistent model features (`layer4.1.gn2`) and (c) more aligned client model parameters (`conv1.weight`).

**Communication cost.**   Communication cost is a critical bottleneck in federated learning (FL), making its optimization an important challenge. In this study, we take FedAvg as the baseline, and define $B$ as the total number of bits exchanged by FedAvg during $T$ training rounds. For each method, we measure its communication cost in terms of (i) the number of rounds required to reach FedAvg's performance, and (ii) the total number of bits exchanged in these rounds. As shown in Table 4, our method significantly reduces both the number of communication rounds and the total transmitted bits compared with state-of-the-art baselines, while achieving comparable performance to FedAvg.

**Training Efficiency.**   Training efficiency is another crucial factor in federated learning (FL), as practical deployments often require methods to reach a target accuracy under limited computational and wall-clock budgets. In this study, we adopt FedAvg as the reference baseline and first run it for 800 rounds on CIFAR-10 under a Dirichlet–0.1 client distribution using a ResNet-18 backbone, recording the highest accuracy it achieves. For each competing method, we then evaluate its training

Table 4: Communication cost comparison on different datasets with ResNet-18. *Note:* Each cell reports **Rounds / Relative Communication Cost (B)**.

| Method | CIFAR-10 | | CIFAR-100 | | Tiny-ImageNet | |
|---|---|---|---|---|---|---|
| | $\alpha = 0.6$ | $\alpha = 0.1$ | $\alpha = 0.6$ | $\alpha = 0.1$ | $\alpha = 0.6$ | $\alpha = 0.1$ |
| FedAvg | 800 / 1.00 | 800 / 1.00 | 800 / 1.00 | 800 / 1.00 | 300 / 1.00 | 300 / 1.00 |
| FedSAM | 465 / 0.58 | 718 / 0.90 | 483 / 0.60 | 491 / 0.61 | 158 / 0.53 | 250 / 0.83 |
| FedSMOO | 205 / 0.51 | 312 / 0.78 | 201 / 0.50 | 225 / 0.56 | 143 / 0.95 | 204 / 1.36 |
| FedGLOSS | 386 / 0.48 | 487 / 0.61 | 261 / 0.33 | 285 / 0.36 | 206 / 0.69 | 264 / 0.88 |
| **FedFFT-D (Ours)** | **190 / 0.24** | **302 / 0.38** | **158 / 0.20** | **211 / 0.26** | **131 / 0.44** | **179 / 0.60** |

efficiency in terms of (i) the number of communication rounds required to match this accuracy level, and (ii) the total wall-clock time consumed. The average per-round time (the "Times" column in Table 5) is computed by dividing the total training time by the number of rounds. As shown in Table 5, although FFT introduces a theoretically higher perturbation cost compared with the linear perturbation used in SAM, its practical overhead is negligible relative to the dominant forward/backward cost. More importantly, the accelerated convergence enabled by our FFT-based optimization leads to a substantial reduction in total wall-clock time. Overall, FedFFT-D achieves the fastest training time among all methods while reaching the same target accuracy as FedAvg.

Table 5: Training wall-clock time to reach the FedAvg accuracy on CIFAR-10 with Dirichlet $\alpha = 0.1$ using Resnet18 as backbone. Experiments run on NVIDIA Tesla A40.

| Method | Times (s) | Rounds | Total Time (s) |
|---|---|---|---|
| FedAvg | 11.73 | 800 | 9390.21 |
| FedSAM | 16.84 | 718 | 12093.24 |
| FedSMOO | 21.74 | 312 | 6785.71 |
| FedGLOSS | 23.07 | 487 | 11235.70 |
| **FedFFT-D (Ours)** | **21.63** | **302** | **6532.98** |

## 6 CONCLUSION

In this work, we introduce a novel frequency-domain perspective to address divergent perturbations in sharpness-aware federated learning. We identify that inter-client disagreements are predominantly a low-frequency phenomenon and accordingly propose FedFFT, a lightweight filtering method to suppress these discordant components. Extensive experiments validate that FedFFT consistently outperforms state-of-the-art methods, particularly in highly non-IID settings, by converging to visibly flatter and wider global minima. This result not only explains the superior generalization and communication efficiency of our method but also establishes spectral analysis as a powerful new tool for designing robust federated optimization algorithms.

## 7 ETHICS STATEMENT

This work adheres to the ICLR Code of Ethics. Our research does not involve human subjects, personally identifiable information, or sensitive data. All datasets used in this work are publicly available and widely used in the research community. We have carefully considered potential risks of misuse, fairness, and bias, and we provide detailed analysis in the experiments to ensure that our methods do not amplify harmful stereotypes or unfair treatment. The results and methodologies are intended solely for academic research and are not designed for deployment in safety-critical or harmful applications.

## 8 REPRODUCIBILITY STATEMENT

We have taken multiple steps to ensure the reproducibility of our results. All details regarding datasets, preprocessing, model architectures, training procedures, and hyperparameters are described

in the Section 5.1 and Appendix B. Complete proofs of theoretical claims are provided in the supplementary materials. To further support reproducibility, we will release the source code and instructions for reproducing all experiments in the supplementary materials.

## 9 LLM USAGE STATEMENT

Large language models (LLMs) were used solely as an assistive tool for language polishing, grammar correction, and improving readability. No part of the research ideation, methodology design, experimental implementation, or result analysis was conducted by LLMs.

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

# A ALGORITHMS

## A.1 FEDFFT-D ALGORITHM.

---

**Algorithm 2** The FedFFT-D Algorithm.

---

**Require:** Communication rounds $T$, local epochs $E$, perturbation radius $\rho$, local learning rate $\eta$, frequency truncation ratio $r$, penalty parameter $\beta$, local multipliers $\{\lambda_k\}_{k=1}^K$, global multiplier $\lambda$

**Ensure:** Global model $w_g^T$

    Initialize global model $w_g^0$, local multipliers $\forall k, \lambda_k = 0$, and global multiplier $\lambda = 0$.

2: **for** $t = 0$ to $T - 1$ **do**

    Randomly select active client set $S_t$.

4:     **for** all clients $k \in S_t$ **in parallel do**

        $w^{k,t,0} \leftarrow w_g^t$                      ▷ Server sends global model to client

6:         **for** $e = 0$ to $E - 1$ **do**

            $\delta^{k,t,e} \leftarrow \rho \cdot \frac{\nabla \mathcal{L}_k(w^{k,t,e})}{\|\nabla \mathcal{L}_k(w^{k,t,e})\|_2}$           ▷ Compute SAM perturbation

8:             $\tilde{\delta}_l^{k,t,e} = \text{Filter}(\delta_l^{k,t,e}, r), \ \forall l \in [1, L]$        ▷ Apply filtering process

            $w^{k,t,e+1} \leftarrow w^{k,t,e} - \eta_l\left(\nabla \mathcal{L}_k(w^{k,t,e} + \tilde{\delta}^{k,t,e}) + \lambda_k + \frac{1}{\beta}(w^{k,t,e} - w_g^t)\right)$    ▷ Dyn

    update step

10:         **end for**

        Send local model $w^{k,t,E}$ to the server.

12:         Update local multiplier: $\lambda_k \leftarrow \lambda_k - \frac{1}{\beta}(w^{k,t,E} - w_g^t)$.

    **end for**

14:     Aggregate models: $w_g^{t+1} \leftarrow w_g^t - \eta_g \sum_{k \in S_t}(w_g^t - w^{k,t,E})$.  ▷ Aggregate models on server

    Update global multiplier: $\lambda \leftarrow \lambda - \frac{1}{\beta|S_t|}\sum_{k \in S_t}(w^{k,t,E} - w_g^t)$.

16: **end for**

    **return** $w_g^T$

---

## A.2 FEDFFT-S ALGORITHM.

---

**Algorithm 3** The FedFFT-S Algorithm.

---

**Require:** Communication rounds $T$, local epochs $E$, perturbation radius $\rho$, local learning rate $\eta$, frequency truncation ratio $r$, penalty parameter $\beta$, local control variates $\{C_k\}_{k=1}^K$, global control variate $C$

**Ensure:** Global model $w_g^T$

    Initialize global model $w_g^0$, local control variates $\forall k, C_k = 0$, and global control variate $C = 0$.

2: **for** $t = 0$ to $T - 1$ **do**

    Randomly select active client set $S_t$.

4:     **for** all clients $k \in S_t$ **in parallel do**

        $w^{k,t,0} \leftarrow w_g^t$                      ▷ Server sends global model to client

6:         **for** $e = 0$ to $E - 1$ **do**

            $\delta^{k,t,e} \leftarrow \rho \cdot \frac{\nabla \mathcal{L}_k(w^{k,t,e})}{\|\nabla \mathcal{L}_k(w^{k,t,e})\|_2}$           ▷ Compute SAM perturbation

8:             $\tilde{\delta}_l^{k,t,e} = \text{Filter}(\delta_l^{k,t,e}, r), \ \forall l \in [1, L]$        ▷ Apply filtering process

            $w^{k,t,e+1} \leftarrow w^{k,t,e} - \eta_l\left(\nabla \mathcal{L}_k(w^{k,t,e} + \tilde{\delta}^{k,t,e}) + C_k - C\right)$   ▷ Scaffold update step

10:         **end for**

        $C_k \leftarrow C_k - C + \frac{1}{\eta E}(w_g^t - w^{k,t,E})$

12:         Send local model $w^{k,t,E}$ and $C_k$ to the server.

    **end for**

14:     Aggregate models: $w_g^{t+1} \leftarrow w_g^t - \eta_g \sum_{k \in S_t}(w_g^t - w^{k,t,E})$.  ▷ Aggregate models on server

    Update global control variate: $C \leftarrow C + \frac{1}{K}C_i$

16: **end for**

    **return** $w_g^T$

---

## B  SAM IN FL

| Research Work | Base Algorithm | Minimizing Target | Perturbation |
|---|---|---|---|
| FedSAM (ECCV22, ICML22) | FedAvg | Local Sharpness | $\rho \cdot \frac{\nabla \mathcal{L}_k(w^{k,t,e})}{\|\nabla \mathcal{L}_k(w^{k,t,e})\|}$ |
| MoFedSAM (ICML22) | FedAvg with Momentum | Local Sharpness | $\rho \cdot \frac{\nabla \mathcal{L}_k(w^{k,t,e})}{\|\nabla \mathcal{L}_k(w^{k,t,e})\|}$ |
| FedGAMMA (TNNLS23) | Scaffold | Local Sharpness | $\rho \cdot \frac{\nabla \mathcal{L}_k(w^{k,t,e})}{\|\nabla \mathcal{L}_k(w^{k,t,e})\|}$ |
| FedSMOO (ICML23) | FedDyn | Local Sharpness with Correction | $\rho \cdot \frac{\nabla \mathcal{L}_k(w^{k,t,e}) - \mu_i - s}{\|\nabla \mathcal{L}_k(w^{k,t,e}) - \mu_i - s\|}$ |
| FedLESAM (ICML24) | FedAvg, Scaffold, FedDyn | Global Sharpness | $\rho \cdot \frac{w_{old}^k - w_g^t}{\|w_{old}^k - w_g^t\|}$ |
| FEDGLOSS (CVPR25) | FedDyn-like | Global Sharpness via Pseudo-gradient | $\rho \cdot \frac{\Delta_w^{t-1}}{\|\Delta_w^{t-1}\|}$ |
| FedFFT (ours) | FedAvg, Scaffold, FedDyn | Local Sharpness with Filtering | $\rho \cdot Filter(\frac{\nabla \mathcal{L}_k(w^{k,t,e})}{\|\nabla \mathcal{L}_k(w^{k,t,e})\|})$ |

Table 6: Summary of federated SAM-based algorithms for solving data heterogeneity.

## C  IMPLEMENTATION OF THE EXPERIMENTS

### C.1  HYPERPARAMETERS

For experiments on CIFAR-10 and CIFAR-100, we adopt the training configurations consistent with FedSMOO (Sun et al., 2023a) and FedLESAM (Fan et al., 2024) for fair comparison. The backbone network is ResNet-18 equipped with Group Normalization and optimized using SGD. The total number of communication rounds is set to 800 for CIFAR-10 and CIFAR-100, and 300 for Tiny ImageNet. The initial local learning rate is $\eta = 0.1$. Unless otherwise specified, the learning rate decays exponentially by a factor of $0.998\times$ per round; however, FedDyn, FedSMOO, FedLESAM-D and FedFFT-D use a slower decay rate of $0.9995\times$ for the proxy term. For CIFAR-10, we use a batch size of 50 and set the number of local epochs to 5. For CIFAR-100, the batch size is 20 with 2 local epochs. For Tiny ImageNet, we follow the same configuration as that of CIFAR-10.

### C.2  MODELS

In our experiments, we adopt different backbone architectures for evaluation. For the experiments reported in Table 1, we use the standard `ResNet18` model from the `torchvision` library, where all Batch Normalization (BN) layers are replaced by Group Normalization (GN) layers to improve training stability in federated settings.

For the experiments in Table 2, we further evaluate three representative architectures: (i) `ResNet20`, implemented following the CIFAR variant with GN layers instead of BN; (ii) `DenseNet121`, where we use `DenseNet_fedlaw` (Li et al., 2023) implementation with GN applied after the final dense block; and (iii) a Vision Transformer, specifically the `vit_tiny_patch16_224` model from the `timm` library. These choices allow us to validate the generality of our approach across both convolutional and transformer-based models.

### C.3  DATASETS

CIFAR-10 and CIFAR-100 are widely used benchmark datasets in computer vision and federated learning research. Both consist of small-scale natural images of size 32×32 with three color channels. CIFAR-10 contains 10 object categories while CIFAR-100 extends this to 100 finer-grained classes, making it a more challenging variant. Despite their limited resolution, these datasets remain popular due to their balanced composition and ease of use in distributed training scenarios.

To further evaluate scalability, Tiny ImageNet is employed, which provides 200 categories of images with higher resolution (64×64). Compared with CIFAR datasets, Tiny ImageNet introduces more diverse classes and larger input dimensions, enabling more comprehensive testing of algorithms under settings with higher model capacity and increased class heterogeneity. Such datasets are especially valuable in federated learning studies, where both efficiency and robustness to distributional challenges are critical.

In addition to image-based benchmarks, we further incorporate the Shakespeare dataset from LEAF, which is constructed from The Complete Works of William Shakespeare (Caldas et al., 2019). In this dataset, each speaking role in each play is treated as a separate device, resulting in a naturally heterogeneous and highly imbalanced client population that closely reflects real-world FL scenarios. Shakespeare contains 1,129 devices with a total of 4,226,158 samples, where the number of samples per device exhibits large variance. To make the experimental setup computationally tractable while preserving the dataset's heterogeneity, we randomly sample 20% of the devices for our experiments. Such pronounced cross-device imbalance and linguistic diversity make Shakespeare a valuable benchmark for evaluating federated learning methods under non-IID text distributions, especially in personalized or robustness-focused settings.

Table 7: Dataset introductions.

| Dataset | Training Data | Test Data | Class Size / Image |
|---|---|---|---|
| CIFAR-10 | 50000 | 10000 | 10 / 3×32×32 |
| CIFAR-100 | 50000 | 10000 | 100 / 3×32×32 |
| Tiny ImageNet | 100000 | 10000 | 200 / 3×64×64 |

### C.4 CLIENT DATA DISTRIBUTION VISUALIZATIONS

In this appendix, we present the client data distributions under different Dirichlet parameters $\alpha$ for CIFAR-10, CIFAR-100, and Tiny ImageNet. Each heatmap shows the number of samples per class for each client.

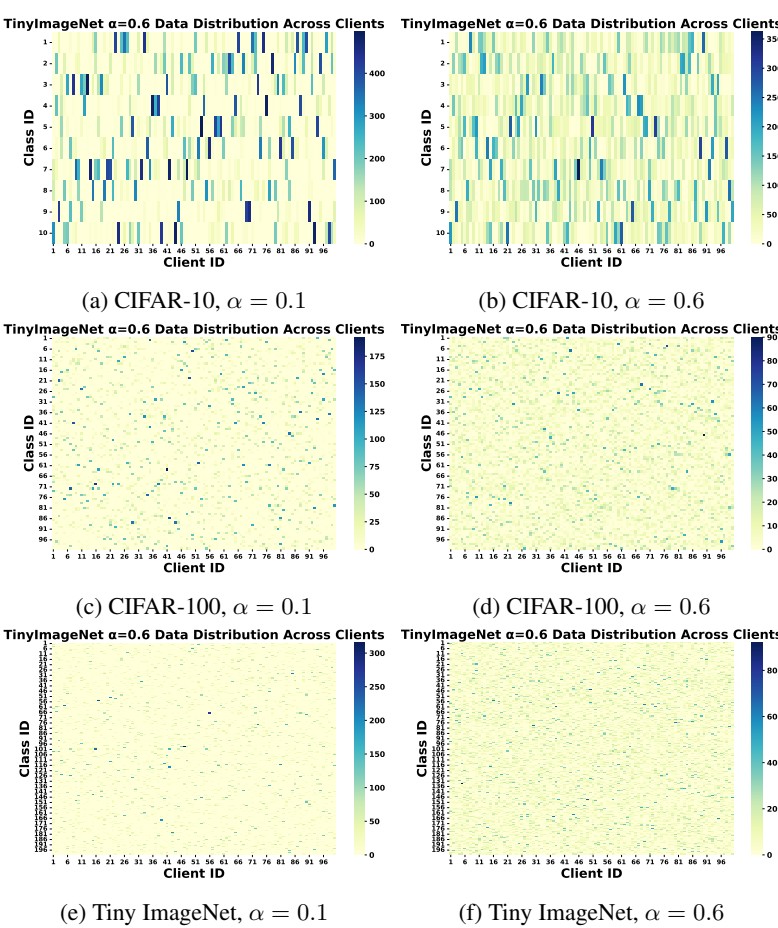

(a) CIFAR-10, $\alpha = 0.1$      (b) CIFAR-10, $\alpha = 0.6$

(c) CIFAR-100, $\alpha = 0.1$      (d) CIFAR-100, $\alpha = 0.6$

(e) Tiny ImageNet, $\alpha = 0.1$      (f) Tiny ImageNet, $\alpha = 0.6$

Figure 5: Client data distributions across different datasets and Dirichlet parameters $\alpha$.

### C.5 NON-IID DATA PARTITIONING WITH DIRICHLET DISTRIBUTION

To simulate realistic data heterogeneity across clients, we partition the training data following a canonical non-IID setup widely used in federated learning literature (Hsu et al., 2019; Li et al., 2021a). Specifically, for a dataset with c classes, we allocate the data of each class to k clients according to a Dirichlet distribution, $Dir_K(\alpha)$, where $\alpha$ is the concentration parameter. A smaller $\alpha$ value induces a more skewed sampling, resulting in each client holding data from only a few classes(high heterogeneity). Conversely, a larger $\alpha$ leads to a more uniform data distribution across-clients (low heterogeneity). In our experiments, we use $\alpha \in \{0.1, 0.6\}$ to simulate high andmoderate heterogeneity regimes, respectively.

## D VISUALIZATION

### D.1 LEARNING CURVE

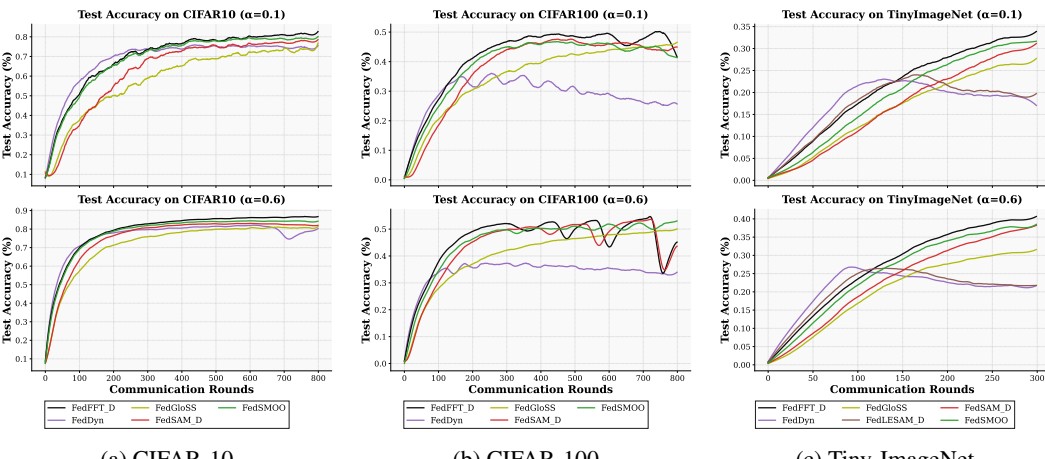

|  |  |  |
|:---:|:---:|:---:|
| (a) CIFAR-10 | (b) CIFAR-100 | (c) Tiny-ImageNet |

Figure 6: Learning curves on three benchmarks.

### D.2 3D LANDSCAPE VISUALIZATION

**Visualization of the Global Loss Landscape.** To visualize the 3D loss landscape, we perturb the parameters of the best-performing checkpoint along its top-two Hessian eigen-directions—computed via power iteration on 500 CIFAR-10 test samples—and plot the corresponding loss values as a smooth surface. As shown in Figure 7, compare to FedSAM, FedSMOO, and FedGloSS, the loss surface corresponding to the FedFFT_D solution is visibly flatter and wider. This provides a clear geometric explanation for our superior generalization performance. By filtering out discordant, client-specific sharpness directions, our method successfully guides the global model to converge not just to a point of low loss, but to a broad, flat minimum that is inherently more robust to the data distribution shifts present across clients. The 2D landscape can be found in the D.3.

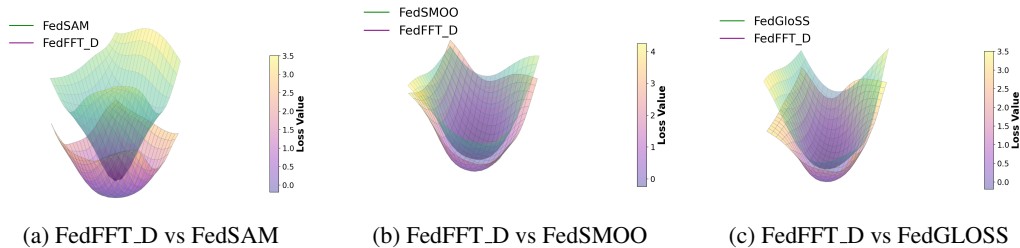

|  |  |  |
|:---:|:---:|:---:|
| (a) FedFFT_D vs FedSAM | (b) FedFFT_D vs FedSMOO | (c) FedFFT_D vs FedGLOSS |

Figure 7: Comparison of 3D loss landscapes on CIFAR-10/ResNet-18 ($\alpha = 0.1$). Visualization of loss landscapes for different federated learning algorithms: (a) FedFFT_D compared with FedSAM, (b) FedFFT_D compared with FedSMOO, (c) FedFFT_D compared with FedGLOSS.

### D.3 2D LANDSCAPE VISUALIZATION

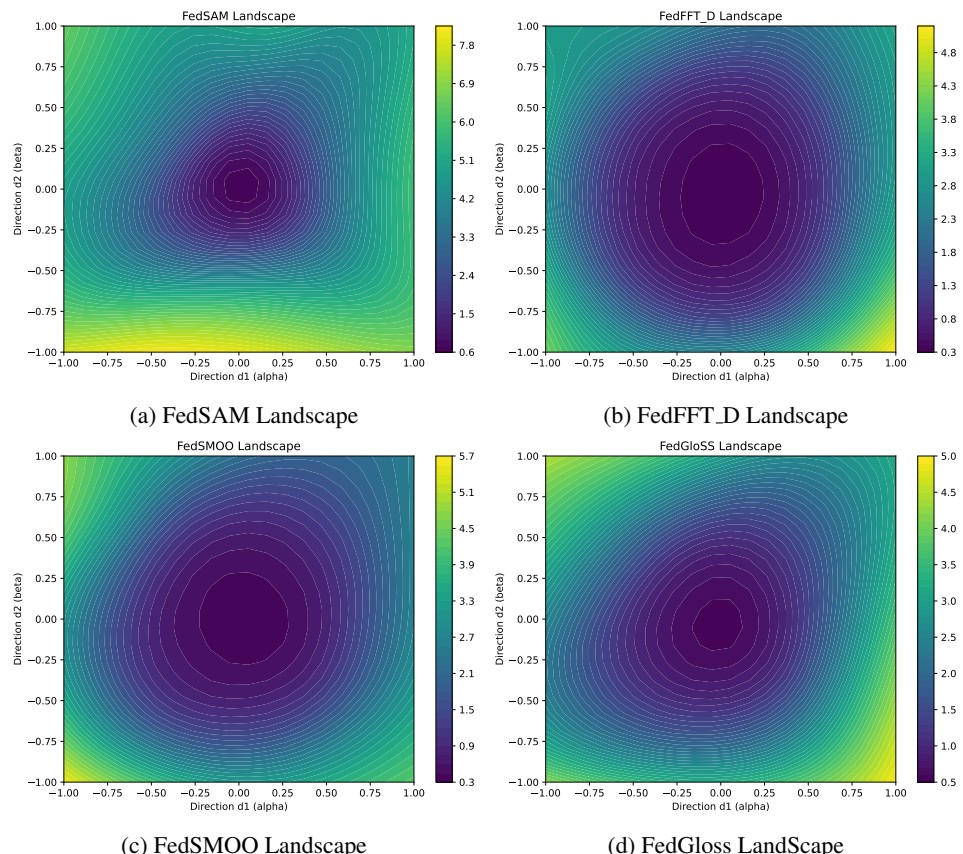

(a) FedSAM Landscape

(b) FedFFT_D Landscape

(c) FedSMOO Landscape

(d) FedGloss LandScape

## E OTHERS

### E.1 DIFFFERENT FILTERING METHODS ON RESNET20.

We further investigate the effect of different spectral filtering strategies on CIFAR-10 using the ResNet-20 backbone, as summarized in Table 8. For a fair comparison, all methods adopt the same filtering ratio of 0.01, i.e., the lowest 1% or highest 1% frequency components are removed, or 1% of frequencies are randomly removed. All methods are implemented based on the FedDyn framework.

Table 8: Performance Comparison of Different Filtering Methods on CIFAR-10 using ResNet-20 backbone.

| Filtering Approach | Accuracy (%) | |
| --- | --- | --- |
| | $\alpha = 0.6$ | $\alpha = 0.1$ |
| FedDyn + SAM | 88.82 | 77.10 |
| High-frequency Filtering | 89.01 | 77.20 |
| Random Filtering | 88.48 | 76.38 |
| Low-frequency Filtering (FedFFT-D) | **91.23** | **81.24** |

### E.2 COMBINE WITH OTHER METHODS.

To further validate the universality of our method, we additionally combine it with FedAWA (Shi et al., 2025), a representative approach that optimizes aggregation weights. We implemented the SAM optimizer on all client sides and used the FedAvg framework for verification on the ResNet-20 model. As shown in Table 9.

Table 9: Test accuracy (%) on CIFAR-10 and CIFAR-100 with Dirichlet distribution parameter $\alpha = 0.1$. The federated learning setup is fixed as **400 rounds, 100 clients, active ratio = 0.1**, and SAM is used as the client optimizer. We compare FedAWA and FedAWA+FedFFT.

| Method | CIFAR-10 | CIFAR-100 |
|---|---|---|
| FedAWA | 47.20 | 33.40 |
| FedAWA+FFT | **50.16** | **34.04** |

### E.3 SENSITIVITY TO THE PERTURBATION RADIUS.

The perturbation radius $\rho$ is an important hyperparameter in SAM-based federated optimization methods. In our main experiments, we follow common practice in the literature and set $\rho = 0.1$, consistent with prior studies such as FedSMOO and FedLESAM. To further examine how the choice of $\rho$ influences performance, we conduct a sensitivity analysis for both FedSAM and our proposed FedFFT under a range of values:

$$\rho \in \{0.05,\ 0.10,\ 0.15,\ 0.20\}.$$

Table 10 summarizes the results on CIFAR-10 with Dirichlet $\alpha = 0.1$ using a ResNet-18 backbone. We observe that FedFFT consistently outperforms FedSAM across all perturbation radii, and the performance gap increases as $\rho$ becomes larger. While FedSAM exhibits substantial degradation when the perturbation radius grows, FedFFT maintains stable accuracy, indicating significantly improved robustness. These findings suggest that the spectral filtering mechanism in FedFFT effectively stabilizes SAM-style perturbations under non-IID data distributions, making our method considerably less sensitive to the choice of $\rho$.

Table 10: Performance comparison between FedSAM and FedFFT under different perturbation radii $\rho$ on CIFAR-10 with Dirichlet $\alpha = 0.1$ using ResNet-18.

| Method | $\rho = 0.05$ | $\rho = 0.10$ | $\rho = 0.15$ | $\rho = 0.20$ |
|---|---|---|---|---|
| FedSAM | 77.91 | 74.92 | 69.38 | 58.49 |
| FedFFT | 78.16 | 77.53 | 75.29 | 71.69 |
| $\Delta$ (Improvement) | +0.25 | +2.61 | +5.91 | +13.20 |

### E.4 BEHAVIOR IN PERSONALIZED FEDERATED LEARNING SETTINGS.

Personalization is an important aspect in federated learning (FL), where methods must retain client-specific information while still benefiting from shared global knowledge. Here, we study how the proposed FedFFT behaves under personalized FL frameworks. In FedPer, where each client maintains a private classification head while sharing the feature extractor. Since FedFFT suppresses high-frequency inconsistencies only within the shared representation—while the personalized head preserves client-unique high-frequency signals—the method does not hinder personalization and can even stabilize the learning of shared features.

To empirically validate this compatibility, we implemented FedPer (Arivazhagan et al., 2019) with a ResNet-18 backbone on the FEMNIST dataset from the LEAF(Caldas et al., 2019) benchmark, a naturally federated dataset featuring substantial client heterogeneity. For each client, we replaced the local optimizer in FedPer with (i) a SAM optimizer and (ii) our FFT-based SAM optimizer, while keeping all other components identical.

As shown in Table 11, FedFFT consistently yields higher accuracy compared to SAM and exhibits smoother convergence behavior. These results confirm that FedFFT effectively preserves client-specific signals while improving the stability of shared representation learning.

Table 11: Comparison between SAM and FFT-SAM optimizers in a personalized FL setting on FEMNIST with ResNet-18.

| Method | Accuracy |
|---|---|
| FedPer + SAM | 65.49 |
| FedPer + **FFT-SAM (ours)** | **67.12** |

### E.5    BEHAVIOR ON NATURAL HETEROGENEITY: SHAKESPEARE (NLP) BENCHMARK

Beyond vision tasks with synthetic Dirichlet partitions, it is important to evaluate whether FedFFT remains effective in more realistic federated environments. To this end, we further study our method on the Shakespeare dataset from LEAF (Caldas et al., 2019), a canonical NLP benchmark for next-character prediction. Unlike artificially partitioned datasets, Shakespeare exhibits naturally occurring non-IID heterogeneity, where each client corresponds to a speaking role with distinct writing styles and vocabulary patterns.

We adopt a two-layer LSTM backbone and implement the standard training pipeline following the publicly available repository `wenzhu23333/Federated-Learning`. Similar to our main experiments, we compare (i) the baseline SAM optimizer used in FedSAM_D, and (ii) our FFT-based SAM optimizer, while keeping all other components identical.

Table 12 reports the results. FedFFT_D consistently outperforms FedSAM_D, confirming that our frequency-domain filtering remains beneficial even in realistic, naturally heterogeneous NLP settings. These findings complement our vision experiments and demonstrate that FedFFT generalizes beyond synthetic Dirichlet setups.

Table 12: Performance on the Shakespeare dataset using a two-layer LSTM.

| Method | Test Accuracy (%) |
|---|---|
| FedSAM_D | 41.49 |
| **FedFFT_D (ours)** | **42.34** |

These results further validate that FedFFT remains effective across modalities (vision, language), model architectures (CNNs, LSTMs), and types of heterogeneity (synthetic Dirichlet, naturally occurring).

### E.6    EFFECT OF APPLYING FREQUENCY FILTERING TO RAW GRADIENTS

An interesting question is whether the proposed frequency-domain filtering would remain effective if applied directly to raw gradients instead of SAM perturbations.

To examine this difference empirically, we constructed a gradient-filtered variant of FedDyn. For each client, we flattened the local gradient, applied rFFT, removed the lowest-frequency components, and reconstructed the filtered gradient via inverse rFFT before performing the update. Experiments were conducted on CIFAR-10 under Dirichlet heterogeneity levels $\alpha = 0.6$ and $\alpha = 0.1$ using a ResNet-18 backbone.

As shown in Table 13 and Figure 9, filtering raw gradients produces nearly identical final accuracy to FedDyn, with only minor benefits in early-stage convergence. These results indicate that the gains introduced by FedFFT stem from filtering *SAM perturbations*, which provide richer geometric information than raw gradients.

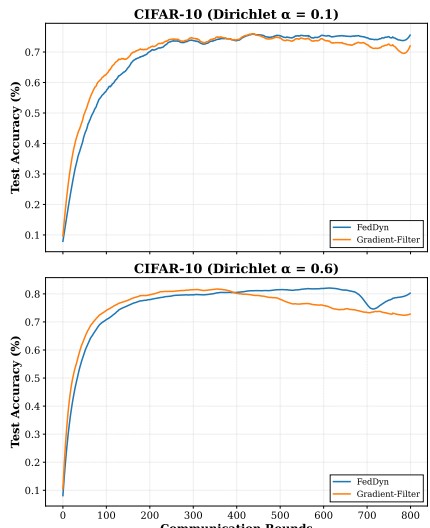

Figure 9: Performance comparison between FedSAM and FedFFT under different perturbation radii $\rho$ on CIFAR-10 with Dirichlet $\alpha = 0.1$ using ResNet-18.

Table 13: Comparison between FedDyn and FedDyn with Gradient Filtering on CIFAR-10.

| Method | $\alpha = 0.6$ | $\alpha = 0.1$ |
|---|---|---|
| FedDyn | 82.62 | 77.61 |
| FedDyn-GradFilter (Ours) | 82.13 | 77.44 |

These observations reinforce the central motivation of FedFFT: filtering frequency components is most effective when applied to geometry-aware SAM perturbations rather than raw gradients, highlighting the importance of local sharpness information for reducing cross-client inconsistency.

## E.7 A FREQUENCY-DOMAIN PERSPECTIVE ON PERTURBATIONS

Our empirical finding that client perturbation disagreements concentrate in the low-frequency spectrum (Figure 1) is not an isolated phenomenon but aligns with broader principles in deep learning. Firstly, it resonates with the concept of spectral bias (Rahaman et al., 2018), which states that neural networks preferentially learn low-frequency functions early in training. This suggests that the most significant, client-specific biases—which fundamentally alter the model's overall function—would naturally manifest as low-frequency components in the parameter updates.

Secondly, treating high-dimensional parameter vectors as signals amenable to frequency analysis has a precedent. For instance, FFT-based gradient sparsification (Wang et al., 2021) successfully leverages the concentration of gradient energy in low-frequencies for communication compression. While (Wang et al., 2021) focuses on the value of gradients for compression, our work focuses on the consistency of SAM perturbations for optimization alignment. This key distinction underscores that we are repurposing a powerful analytical tool to address a different core challenge in federated learning—client drift. By filtering out the discordant low-frequency components of the perturbations, FedFFT directly targets the macroscopic misalignments between clients, guiding them towards a more consistent and robust global solution.

