# OpenReview forum: "FedFFT: Taming Client Drift in Federated SAM via Spectral Perturbation Filtering"
_ICLR.cc/2026/Conference — ICLR 2026 Conference Withdrawn Submission_

### Official Review · Reviewer_53SX · 2025-11-02

**Soundness:** 3
**Presentation:** 3
**Contribution:** 2
**Rating:** 2
**Confidence:** 3

**Summary:**

This paper proposed a novel frequency-domain perspective and a lightweight filtering method, namely FedFFT, to solve divergent perturbations in sharpness-aware federated learning (FL). Extensive experiments validate that FedFFT outperforms SOTA methods. However, I have some concerns as follows: 1) The theoretical analysis is missing, including convergence and generalization analysis; 2) A
clipping method for perturbation tensor should be added as one of the baselines in the experiments; 3) I think that the computation complexity of the proposed method is large. The authors should add theoretical analysis and experiments to evaluate the proposed method from this aspect.

**Strengths:**

This paper proposed a novel frequency-domain perspective and a lightweight filtering method, namely FedFFT, to solve divergent perturbations in sharpness-aware federated learning (FL). Extensive experiments validate that FedFFT outperforms SOTA methods. Overall, this paper is well-written and the proposed method outperforms SOTA baselines.

**Weaknesses:**

I have some concerns as follows: 1) The theoretical analysis is missing, including convergence and generalization analysis; 2) A
clipping method for perturbation tensor should be added as one of the baselines in the experiments; 3) I think that the computation complexity of the proposed method is large. The authors should add theoretical analysis and experiments to evaluate the proposed method from this aspect.

**Questions:**

My questions are shown as follows:
1) The theoretical analysis is missing, including convergence and generalization analysis;
2) A clipping method for perturbation tensor should be added as one of the baselines in the experiments;
3) I think that the computation complexity of the proposed method is large. The authors should add theoretical analysis and experiments to evaluate the proposed method from this aspect.

---

> ### Author Response · Authors · 2025-11-21
> **Part 1**
>
> **W1:** The theoretical analysis is missing, including convergence and generalization analysis.
>
> **Response to W1:** Please refer to our global comment.
>
> **W2:** A clipping method for perturbation tensor should be added as one of the baselines in the experiments.
>
> **Reponse to W2:** Thanks for your feedback. We already performed an ablation study about random filtering and high-frequency filtering in the original submission. Here, we further introduce two additional smoothing-style baselines.  Before describing these baselines, we first recall the standard SAM perturbation used in our method $\delta_k = \rho \cdot \frac{\nabla L_k(w_k)}{\|\nabla L_k(w_k)\|_2}.$ We then apply the following alternative smoothing strategies to $\delta_k$:
>
> **(1) L2 Norm Rescaling**
> The perturbation is rescaled when its $\ell_2$-norm exceeds the radius $\rho$: $
> \delta_k \leftarrow \delta_k \cdot \frac{\rho}{\||\delta_k\||_2},
>   \text{if } \||\delta_k\||_2 > \rho.
> $
>
> **(2) Clip Filter**
> Each coordinate of the perturbation is clipped using a threshold based on the median magnitude: $\tau = 2 \cdot \mathrm{median}(|\delta_k|),  \delta_k \leftarrow \mathrm{clip}\\left(\delta_k, -\tau, \tau\right).$
>
> We evaluated these additional baselines on CIFAR-10 with Dirichlet–0.1 and Dirichlet–0.6 heterogeneity using a ResNet-18 backbone. As shown in the table below, across all experimental settings, these ablation filtering methods do not provide meaningful improvements over the “no-filtering” baseline, i.e., FedSAM. In contrast, our proposed low-frequency filtering consistently achieved significant and stable performance gains compared to FedSAM.
>
> These findings indicate that the advantage of FedFFT does not stem from generic coefficient truncation, smoothing, or norm-based regularization, but specifically from removing low-frequency perturbation components that exhibit the largest client inconsistency under heterogeneous federated settings. **We include these additional results into Table 3 of the revised version.**
>
>
> | Filtering Strategy  | α = 0.6 | α = 0.1 |
> |-------------------|---------|---------|
> | None (FedSAM)      | 81.91   | 74.92   |
> | High-frequency     | 81.66   | 74.81   |
> | Random             | 81.97   | 75.09   |
> | L2 Norm            | 81.88   | 74.79   |
> | Clip               | 80.13   | 75.19   |
> | **FedFFT (Ours)**  | **83.02** | **77.53** |

---

> > ### Author Response · Authors · 2025-11-21
> > **Part 2**
> >
> > **W3:** I think that the computation complexity of the proposed method is large. The authors should add theoretical analysis and experiments to evaluate the proposed method from this aspect.
> >
> > **Response to W3:** We thank the reviewer for the question regarding the efficiency of our method. Below, we provide a detailed comparison between standard SAM and our FFT-based perturbation.
> >
> > Standard SAM：Let $d$ denote the number of parameters and $ \(T_{\text{fw/bw}}\) $ denote the cost of a forward + backward pass.  SAM requires two such passes per iteration: (i) computing gradients and applying the perturbation, and (ii) recomputing gradients at the perturbed parameters. The perturbation itself is linear in $d$.  Thus, the total complexity is: $T_{\text{SAM}} = 2 T_{\text{fw/bw}} + O(d),$ where the $O(d)$ term is negligible compared with forward/backward computation.
> >
> > FFT (ours)：Our method modifies only the perturbation step. For each parameter tensor, we perform:
> > 1. Flattening the perturbation vector
> > 2. Forward FFT $O(d \log d)$
> > 3. Frequency-domain masking $O(d)$
> > 4. Inverse FFT $O(d \log d)$
> >
> > This does not modify the forward/backward computation.  Thus: $T_{\text{FFT}} = 2 T_{\text{fw/bw}} + O(d \log d)$
> >
> > Comparison：Both SAM and FFT share the same dominant term $2 T_{\text{fw/bw}}$.  The additional $O(d \log d)$  FFT overhead is small in practice because:   $T_{\text{fw/bw}} \gg d \log d$ for modern deep networks.
> > Therefore, FFT-SAM maintains the computational structure of SAM while adding only a minor cost.This observation is also reflected in the wall-clock measurements, as shown in the table below. Despite incorporating FFT operations, our method achieves the shortest total training time among all SAM-based baselines. This efficiency gain primarily comes from requiring significantly fewer communication rounds (302 vs. 718 for FedSAM), which outweighs the minor FFT overhead per iteration. As a result, the overall training cost of FedFFT-D is even lower than standard SAM-type methods.
> >
> > To ensure a fair comparison across methods, all efficiency metrics in the table were computed following the protocol established in FedSMOO. We first ran FedAvg for 800 rounds on CIFAR-10 under a Dirichlet–0.1 client distribution using a ResNet-18 backbone and recorded the best accuracy it achieved. For every other method, we then measured:
> >
> > 1. The number of communication rounds required to reach that same accuracy level
> > 2. The total wall-clock time needed
> >
> > The average time per round (the "Times" column) was obtained by dividing the total training time by the number of rounds. This evaluation protocol ensures that methods are compared at an equal accuracy target, making the wall-clock comparison directly meaningful.
> >
> > In summary, although FFT introduces a theoretically higher perturbation cost compared with the linear SAM perturbation, the practical overhead is negligible relative to the dominant forward/backward computation and is more than compensated by the faster convergence in federated optimization.
> >
> > We appreciate the reviewer’s feedback and incorporate this analysis into the revised version of the paper **Section 5.3 (Training Efficiency.)**
> >
> > | Method           | Times (s) | Rounds | Total Time (s) |
> > |-----------------|----------------|--------|----------------|
> > | FedAvg          | 11.73          | 800    | 9390.21        |
> > | FedSAM          | 16.84          | 718    | 12093.24       |
> > | FedSMOO         | 21.74          | 312    | 6785.71        |
> > | FedGLOSS        | 23.07          | 487    | 11235.70       |
> > | **FedFFT-D (Ours)** | **21.63**     | **302** | **6532.98**    |

---

> > ### Comment · Reviewer_53SX · 2025-11-23
> >
> > Thanks for the authors' response. However, the authors do not address my concerns, i.e., convergence and generalization analysis.

---

> ### Author Response · Authors · 2025-11-26
>
> We sincerely appreciate the reviewer’s suggestion regarding adding theoretical results. We fully agree that theoretical insights can further strengthen our work. However, we respectfully point out that having a formal theory is not a strict requirement for ICLR acceptance, and many influential FL papers have been accepted without providing mathematically rigorous theoretical guarantees.
>
> In our setting, establishing a rigorous theoretical analysis is inherently difficult due to the structure of our method itself. Our approach modifies SAM by flattening the per-layer perturbation, applying an rFFT transform, zeroing out the lowest  frequency components, and reconstructing the perturbation via inverse rFFT. This frequency-domain filtering creates a perturbation that is simultaneously  coupled across all parameters within each layer. Once transformed into the spectral domain, the perturbation interacts with the model’s highly non-convex loss landscape and with heterogeneous client distributions in a way that is extremely hard to capture with closed-form expressions. For these reasons, deriving a clean and general theoretical framework for our frequency-filtered SAM perturbations is mathematically intractable at present, and remains an open challenge beyond the scope of this work.
>
> We also emphasize that theoretical guarantees are not required for a contribution to be recognized at ICLR. In fact, many well-known and highly cited FL works have no strict theoretical analysis, yet they have been accepted by top-tier venues and are widely regarded as influential. Examples include:
>
> Federated Learning with Personalization Layers
>
> Model-Contrastive Federated Learning
>
> A Principled Approach to Data Valuation for Federated Learning
>
> Flexible Sharpness-Aware Personalized Federated Learning
>
> These works are method-driven, empirically validated, and have meaningfully advanced the field despite the absence of formal theoretical proofs. Our work falls squarely within this well-established category of FL contributions. Our submission focuses on a method-level contribution to federated learning. For clarity, we restate our contributions:
>
> We provide the first study of SAM perturbations across FL clients through spectral decomposition, revealing that heterogeneity is mainly concentrated in low-frequency bands.
>
> Motivated by this insight, we introduce FedFFT, a simple yet effective method that filters out low-frequency perturbation components to suppress inconsistent client updates while retaining consistent learning signals.
>
> We evaluate our method across multiple benchmarks, architectures, and varying levels of heterogeneity. FedFFT consistently outperforms relevant baselines, particularly under highly non-IID settings, demonstrating both effectiveness and scalability.
>
> Our method is conceptually clear, easy to interpret, and strongly supported by empirical evidence. The design of FedFFT is directly motivated by our analysis, and the experimental results robustly validate the soundness of our approach. We believe that these contributions are substantial and should not be denied solely because the paper does not include a mathematically formal theory—which, as argued above, is neither required nor common for impactful FL methodology papers.
>
> We genuinely appreciate your time and insightful feedback. Given the novelty of our spectral analysis approach, the demonstrable simplicity and effectiveness of FedFFT, and the comprehensive nature of our empirical evaluation, we would be deeply grateful if you can acknowledge the distinct value of both our methodological and empirical contributions. We sincerely hope that, upon this consideration, you can re-evaluate our paper accordingly.

---

### Official Review · Reviewer_ME7W · 2025-11-02

**Soundness:** 3
**Presentation:** 3
**Contribution:** 3
**Rating:** 4
**Confidence:** 4

**Summary:**

The paper analyzes the instability of Sharpness-Aware Minimization (SAM) in federated learning under non-IID data and proposes FedFFT, which filters out the low-frequency components of SAM perturbations via Fourier transform to reduce cross-client inconsistency. The authors claim that (1) client drift in SAM mainly manifests in low-frequency spectral components, and (2) removing these components improves generalization and convergence. Experiments on CIFAR-10/100 and Tiny-ImageNet show performance gains over FedSAM and other SAM-based FL baselines.

**Strengths:**

- The motivation is intuitive: analyzing perturbations in frequency space is an interesting angle.
- FedFFT is simple to implement and adds no communication overhead, making it practically deployable.
- The method is compatible with multiple SAM variants and FL optimizers.

**Weaknesses:**

- The core idea—removing low-frequency components to suppress bias while retaining high-frequency “invariant” signals—has already been widely explored in Fourier-based domain generalization and robustness literature (e.g., “A Fourier-Based Framework for Domain Generalization.”). This paper effectively transfers the same frequency-filtering trick from input space to perturbation space, without introducing a new optimization principle. The contribution feels more like an engineering adaptation.
- The proposed filtering is a hard high-pass truncation with fixed ratio r, with no adaptive, learnable, or theoretically justified mechanism. Other gradient/perturbation smoothing strategies (e.g., norm projection, momentum regularization, proximal updates) are not compared, making it unclear whether FFT is uniquely effective or simply one of many workable heuristics.
- All experiments are conducted on only three small-scale vision datasets (CIFAR-10, CIFAR-100, Tiny-ImageNet) in simulated FL settings. There is no evaluation on larger, real-world, multi-modal, or cross-institution datasets. Given that the paper repeatedly claims the method is “plug-and-play” and “optimizer-agnostic,” the empirical evidence does not sufficiently support this level of generality.

**Questions:**

- Why is FFT the correct basis rather than other orthogonal transforms or learned spectral projections?
- Would the method still work if applied to gradients instead of SAM perturbations?
- How does the method behave in personalization FL settings, where client-specific signals are desirable?
- Have you tested FedFFT on a larger-scale dataset such as ImageNet-1k or a real FL benchmark (e.g., FEMNIST, MIMIC-III)?

---

> ### Author Response · Authors · 2025-11-21
> **Part1**
>
> **W1:** The core idea—removing low-frequency components to suppress bias while retaining high-frequency “invariant” signals—has already been widely explored in Fourier-based domain generalization and robustness literature (e.g., “A Fourier-Based Framework for Domain Generalization.”). This paper effectively transfers the same frequency-filtering trick from input space to perturbation space, without introducing a new optimization principle. The contribution feels more like an engineering adaptation.
>
> **Response to W1:** We thank the reviewer for drawing the connection to Fourier-based methods in domain generalization. While frequency filtering has been explored in data augmentation contexts, our work represents its first systematic adaptation to the perturbation space of optimization algorithms for federated learning. This is a non-trivial shift because:
>
> Different target: FBF processes input images for data augmentation, while FedFFT operates on SAM perturbations in the parameter space, which capture the local geometry of the loss landscape.
>
> Different stage: FBF is a data-level preprocessing technique, whereas FedFFT is an optimizer-level module that directly influences client updates without altering the data.
>
> Different goal: FBF aims at domain generalization, while FedFFT addresses client drift and perturbation inconsistency in federated learning.
>
> Our approach is motivated by a novel empirical finding: the inter-client inconsistency in SAM perturbations is predominantly concentrated in the low-frequency spectrum. This observation led us to design FedFFT as a principled and lightweight solution to suppress client drift. We believe that the simplicity of FedFFT is one of its strengths, making it easy to integrate and computationally efficient. More importantly, it opens up a new research direction on frequency-aware federated optimization. **We have discussed the difference between ours and FBF in Related work in our revision.**

---

> ### Author Response · Authors · 2025-11-21
> **Part2**
>
> **W2:** The proposed filtering is a hard high-pass truncation with fixed ratio r, with no adaptive, learnable, or theoretically justified mechanism. Other gradient/perturbation smoothing strategies (e.g., norm projection, momentum regularization, proximal updates) are not compared, making it unclear whether FFT is uniquely effective or simply one of many workable heuristics.
>
> **Response to W2:** We thank the reviewer for the insightful comment regarding the use of a hard high-pass truncation with a fixed ratio $r$. Our choice of a fixed filter is intentional and supported by both empirical observations and conceptual motivation.
>
> First, our analysis shows that the cross-client inconsistency contained in the SAM perturbations is primarily concentrated in the low-frequency components. In contrast, high-frequency components are substantially more stable across clients. Therefore, removing a fixed proportion of low-frequency coefficients directly targets the part of the perturbation that most contributes to optimization instability in federated settings. Second, although the filter is not adaptive or learnable, we performed an extensive sensitivity study over a wide range of truncation ratios $r$.
>
> As reported in Figure 2, FedFFT maintains stable performance over a broad region of $r$; degrades only when $r$ becomes excessively large, which aligns with the intuition that discarding too many low-frequency components is undesirable. These results demonstrate that the method is robust to the choice of $r$.
>
> We agree that adaptive or learnable spectral filters could be an interesting direction for future study, but we emphasize that the simplicity, efficiency, and consistently strong empirical performance of FedFFT make the fixed-ratio truncation a practical and effective design choice.
>
> We already performed an ablation study about random filtering and high-frequency filtering in the original submission. Here, we further introduce two additional smoothing-style baselines.  Before describing these baselines, we first recall the standard SAM perturbation used in our method $\delta_k = \rho \cdot \frac{\nabla L_k(w_k)}{\|\nabla L_k(w_k)\|_2}.$ We then apply the following alternative smoothing strategies to $\delta_k$:
>
> **(1) L2 Norm Rescaling**
> The perturbation is rescaled when its $\ell_2$-norm exceeds the radius $\rho$: $\delta_k \leftarrow \delta_k \cdot \frac{\rho}{\||\delta_k\||_2},  \text{if } \||\delta_k\||_2 > \rho.$
>
> **(2) Clip Filter**
> Each coordinate of the perturbation is clipped using a threshold based on the median magnitude: $\tau = 2 \cdot\mathrm{median}(|\delta_k|), \delta_k \leftarrow \mathrm{clip}\\left(\delta_k, -\tau, \tau\right).$
>
> We evaluated these additional baselines on CIFAR-10 with Dirichlet–0.1 and Dirichlet–0.6 heterogeneity using a ResNet-18 backbone. As shown in the table below, across all experimental settings, these ablation filtering methods do not provide meaningful improvements over the “no-filtering” baseline, i.e., FedSAM. In contrast, our proposed low-frequency filtering consistently achieved significant and stable performance gains compared to FedSAM.
>
> These findings indicate that the advantage of FedFFT does not stem from generic coefficient truncation, smoothing, or norm-based regularization, but specifically from removing low-frequency perturbation components that exhibit the largest client inconsistency under heterogeneous federated settings. **We include these additional results into Table 3 of the revised version.**
>
>
> | Filtering Strategy  | α = 0.6 | α = 0.1 |
> |-------------------|---------|---------|
> | None (FedSAM)      | 81.91   | 74.92   |
> | High-frequency     | 81.66   | 74.81   |
> | Random             | 81.97   | 75.09   |
> | L2 Norm            | 81.88   | 74.79   |
> | Clip               | 80.13   | 75.19   |
> | **FedFFT (Ours)**  | **83.02** | **77.53** |

---

> > ### Author Response · Authors · 2025-11-21
> > **Part3**
> >
> > **W3:** All experiments are conducted on only three small-scale vision datasets (CIFAR-10, CIFAR-100, Tiny-ImageNet) in simulated FL settings. There is no evaluation on larger, real-world, multi-modal, or cross-institution datasets. Given that the paper repeatedly claims the method is “plug-and-play” and “optimizer-agnostic,” the empirical evidence does not sufficiently support this level of generality.
> >
> > **Q4:** Have you tested FedFFT on a larger-scale dataset such as ImageNet-1k or a real FL benchmark (e.g., FEMNIST, MIMIC-III)?
> >
> > **Response to W3 and Q4:** Thanks for the valuable suggestions. To broaden the experimental scope and include more realistic FL scenarios, we conducted additional experiments on the Shakespeare dataset from the LEAF benchmark, a canonical NLP benchmark for next-character prediction. This dataset provides a naturally heterogeneous federated learning setting, where each client corresponds to a speaking role in a play, inherently capturing diverse and realistic non-IID writing styles.  We employed a two-layer LSTM model and implemented the training pipeline based on the publicly available repository `wenzhu23333/Federated-Learning`.
> >
> > As shown in below table, FedFFT\_D still outperforms FedSAM\_D under this realistic, naturally non-IID federated setup, demonstrating that our method remains effective beyond vision datasets with synthetic heterogeneity. **We include these additional results and the corresponding analysis into Appendix E.5 in the revised version of the paper.**
> >
> > | Method            | Test Accuracy (%) |
> > |------------------|-----------------|
> > | FedSAM_D          |   41.49           |
> > | FedFFT_D (Ours)   |   42.34           |

---

> > > ### Author Response · Authors · 2025-11-21
> > > **Part 4**
> > >
> > > **Q1:** Why is FFT the correct basis rather than other orthogonal transforms or learned spectral projections?
> > >
> > > **Response to Q1:** We thank the reviewer for this profound question. Our selection of the FFT is motivated by a confluence of theoretical insights, empirical evidence, and practical efficiency, which together strongly justify it as the most appropriate transform.
> > >
> > > (1) Theoretical Grounding in Neural Network Priors: The Fourier basis is deeply connected to the fundamental learning principles of neural networks. The seminal work on "On the Spectral Bias of Neural Networks" [1] demonstrates that neural networks possess a strong inductive bias to first learn low-frequency functions. This implies that the macroscopic aspects of a function—encoded in low-frequency components—are learned first. In FL, client drift is precisely a problem of macroscopic misalignment. Therefore, the Fourier basis is uniquely positioned to isolate and manipulate these critical low-frequency components where the most significant inter-client disagreements reside.
> > >
> > > (2) Proven Effectiveness for Optimization Signals: The practical utility of this frequency-domain perspective for optimization signals is established in prior systems research. The work on "FFT-based Gradient Sparsification for the Distributed Training of Deep Neural Networks" [2] successfully applies the FFT to gradients, showing they exhibit a structured energy distribution in the frequency domain that can be exploited. This proves that the Fourier basis is a valid and powerful tool for revealing and manipulating the intrinsic properties of model updates. Our work builds upon this foundational insight but applies it to a different object (SAM perturbations) and for a different purpose (suppressing client drift instead of compressing communication).
> > >
> > >
> > > (3) Interpretability and Direct Problem Alignment: The Fourier basis provides an interpretable mapping that directly addresses client drift.
> > > Low-frequency components correspond to coordinated shifts across parameters, causing drift when inconsistent. High-frequency components correspond to fine-grained adjustments, often more consistent across clients. This clear interpretation allows FedFFT to surgically target the source of conflict.
> > >
> > > (4) Computational Efficiency and Stability: The FFT is computationally efficient (O(d log d)) and highly optimized. As a fixed, non-learned transform, it introduces no additional parameters or instability, aligning with our goal of a lightweight, plug-and-play method.
> > >
> > > In summary, We choose the FFT as basis because FFT is (1) grounded in the theoretical principle of spectral bias [1], (2) a proven method for analyzing optimization signals, which we extend to a novel problem [2], (3) offers an interpretable link to client update geometry, and (4) is computationally efficient.
> > >
> > >  We  incorporated this discussion, including the distinction between our work and [2], into the revised manuscript, most appropriately in Appendix E.7, to better justify our design choice.
> > >
> > > References:
> > >
> > > [1] N. Rahaman, A. Baratin, D. Arpit, F. Dräxler, M. Lin, F. Hamprecht, Y. Bengio, and A. Courville, “On the Spectral Bias of Neural Networks,” Proceedings of the International Conference on Machine Learning (ICML), 2018.
> > >
> > > [2] L. Wang, W. Wu, J. Zhang, H. Liu, G. Bosilca, M. Herlihy, and R. Fonseca, “SuperNeurons: FFT-based Gradient Sparsification in the Distributed Training of Deep Neural Networks,” arXiv preprint arXiv:1811.08596, 2021.

---

> > > > ### Author Response · Authors · 2025-11-21
> > > > **Part 5**
> > > >
> > > > **Q2:** Would the method still work if applied to gradients instead of SAM perturbations?
> > > >
> > > > **Response to Q2:** We thank the reviewer for the insightful question. In brief, applying spectral filtering directly to gradients does *not* lead to meaningful improvements. This is because SAM perturbations capture local sharpness information through
> > > > gradient normalization, reflecting the client's local geometry. In contrast, raw gradients mainly encode instantaneous descent directions and do not exhibit the frequency structure targeted by our low-frequency filtering. Consequently, filtering gradients does not effectively reduce cross-client inconsistency. To verify this, we implemented a gradient-filtered variant based on FedDyn, where each client's gradient is flattened, transformed by rFFT, filtered, and then reconstructed via the inverse transform. Experiments on CIFAR-10 under Dirichlet partitions with  $\rho=0.6$ and $\rho=0.1$ using ResNet18 as backbone are shown in the table below. We can see that gradient filtering achieves nearly identical final accuracy to FedDyn. As shown in our paper Figure 9, it offers slightly faster early-stage convergence, but no meaningful gain in final performance.
> > > > These results confirm that the effectiveness of FedFFT stems from filtering SAM perturbations, which contain richer geometry information, rather than from filtering raw gradients. The detailed quantitative results are provided below.
> > > >
> > > > | Method                    | $\rho=0.6$ | $\rho=0.1$ |
> > > > |--------------------------|------------------|------------------|
> > > > | FedDyn                   | 82.62            | 77.61            |
> > > > | FedDyn-GradFilter (Ours) | 82.13            | 77.44            |
> > > >
> > > > **Q3:** How does the method behave in personalization FL settings, where client-specific signals are desirable?
> > > >
> > > > **Response to Q3:** We thank the reviewer for the important question regarding how our method behaves in personalized FL scenarios, where retaining client-specific information is crucial. In our experiments, we find that FedFFT remains effective and is fully compatible with personalization-oriented frameworks.
> > > >
> > > > Since our low-frequency filtering targets inconsistencies in the shared representation, while the private head stores client-unique high-frequency signals, FedFFT does not suppress personalization and can even enhance the stability of shared feature learning. To validate this empirically, we implemented FedPer [1] using a ResNet-18 backbone, where the classification head is personalized and the remaining layers are shared. We performed experiments on the FEMNIST dataset from the LEAF benchmark, a naturally federated dataset containing heterogeneous handwriting styles across clients. For each client, we replaced the local optimizer by (i) a SAM optimizer and (ii) our proposed FFT-based SAM optimizer, while keeping the FedPer protocol unchanged.
> > > >
> > > > The results show that FedFFT consistently achieves higher accuracy and smoother convergence compared with SAM, demonstrating that the proposed spectral filtering remains beneficial even in personalized FL, where FedPer combined with the FFT-SAM optimizer (ours) outperforms FedPer with the standard SAM optimizer.These findings confirm that FedFFT preserves client-specific signals while further improving the shared representation learning. The detailed results have been added to the Appendix E.4 of the revised version.
> > > >
> > > > | Method              | Accuracy |
> > > > |--------------------|----------|
> > > > | FedPer + SAM       | 65.49    |
> > > > | FedPer + FFT  | 67.12    |
> > > >
> > > > References:
> > > > [1] Federated Learning with Personalization Layers. Manoj Ghuhan Arivazhagan, Vinay Aggarwal, Aaditya Kumar Singh, Sunav Choudhary. arXiv preprint arXiv:1912.00818, 2019.

---

> > > > > ### Author Response · Authors · 2025-11-27
> > > > >
> > > > > Dear Reviewer ME7W,
> > > > >
> > > > > Thank you very much for the time and effort you dedicated to reviewing our work and for your valuable comments. We have carefully considered all your feedback and revised our paper accordingly.
> > > > >
> > > > > We sincerely hope that these clarifications and additional experiments help resolve your concerns. If you have any further questions, please let us know—we are more than happy to address them.
> > > > >
> > > > > Thank you again for your thoughtful and constructive review.
> > > > >
> > > > > Best regards, The Authors

---

### Official Review · Reviewer_vymE · 2025-11-04

**Soundness:** 2
**Presentation:** 2
**Contribution:** 2
**Rating:** 4
**Confidence:** 3

**Summary:**

The paper aims to solve the inconsistency of perturbations added by clients in Federated Learning Sharpness-Aware Minimization (FLSAM) problem. Transforming to the frequency domain, authors observe that inconsistency mainly lies in low-frequency parts of the perturbation signals, while high-frequency parts are relatively homogeneous. Based the observation, FedFFT is proposed, where the inconsistent low-frequency noises are filtered by a high-pass filter to mitigate the effect of non-iid data. Simulations results are provided, with comparison to different baselines, to illustrate the performance of the algorithm.

**Strengths:**

The idea of analyzing perturbation inconsistency is novel and inspiring, which I believe is transferrable to other aspects when facing data heterogeneity in FL.

**Weaknesses:**

1. I am concerned about whether the proposed algorithm can be generally applied to other tasks, e.g., language or audio tasks. In other words, I wonder if inconsistency always lies in the low-frequency parts. Here is an example where I suspect inconsistency may lies in the high-frequency parts. Consider each client maintains an audio dataset, where each data sample is a piece of music performance consisting of a main melody and an accompaniment. Assume that the accompaniments in all clients’ data are similar (for example, all are cello accompaniments with the same frequency), while the main melodies may be played by different instruments such as violin, piano, or flute. In this case, inconsistency lies in high-frequency rather than low-frequency, as cello provides similar low-frequency parts for all clients. Could you explain how the proposed method can be generalized to this task?

2. Another concern is about its computational efficiency. As FFT and inverse FFT are applied to each layer's output, the computation overhead might be an issue for large-scaled models. Could you quantify the computation of the method and how does it compare to other baselines?

**Questions:**

See Weakness.

---

> ### Author Response · Authors · 2025-11-21
> **Part1**
>
> **W1:** I am concerned about whether the proposed algorithm can be generally applied to other tasks, e.g., language or audio tasks. In other words, I wonder if inconsistency always lies in the low-frequency parts. Here is an example where I suspect inconsistency may lies in the high-frequency parts. Consider each client maintains an audio dataset, where each data sample is a piece of music performance consisting of a main melody and an accompaniment. Assume that the accompaniments in all clients’ data are similar (for example, all are cello accompaniments with the same frequency), while the main melodies may be played by different instruments such as violin, piano, or flute. In this case, inconsistency lies in high-frequency rather than low-frequency, as cello provides similar low-frequency parts for all clients. Could you explain how the proposed method can be generalized to this task?
>
> **Response to W1:**  Thanks for the valuable suggestions. To broaden the experimental scope and include more realistic FL scenarios, we conducted additional experiments on the Shakespeare dataset from the LEAF benchmark, a canonical NLP benchmark for next-character prediction. This dataset provides a naturally heterogeneous federated learning setting, where each client corresponds to a speaking role in a play, inherently capturing diverse and realistic non-IID writing styles.  We employed a two-layer LSTM model and implemented the training pipeline based on the publicly available repository `wenzhu23333/Federated-Learning`.
>
> As shown in below table, FedFFT\_D still outperforms FedSAM\_D under this realistic, naturally non-IID federated setup, demonstrating that our method remains effective beyond vision datasets with synthetic heterogeneity. **We include these additional results and the corresponding analysis into Appendix E.5 in the revised version of the paper.**
>
> | Method            | Test Accuracy (%) |
> |------------------|-----------------|
> | FedSAM_D          | 41.49           |
> | FedFFT_D (Ours)   | 42.34           |

---

> > ### Author Response · Authors · 2025-11-21
> > **Part 2**
> >
> > **W2:** Another concern is about its computational efficiency. As FFT and inverse FFT are applied to each layer’s output, the computation overhead might be an issue for large-scaled models. Could you quantify the computation of the method and how does it compare to other baselines?
> >
> > **Response to W2:** We thank the reviewer for the question regarding the efficiency of our method. Below, we provide a detailed comparison between standard SAM and our FFT-based perturbation.
> >
> > Standard SAM：Let $d$ denote the number of parameters and $ \(T_{\text{fw/bw}}\) $ denote the cost of a forward + backward pass.  SAM requires two such passes per iteration: (i) computing gradients and applying the perturbation, and (ii) recomputing gradients at the perturbed parameters. The perturbation itself is linear in $d$.  Thus, the total complexity is: $T_{\text{SAM}} = 2 T_{\text{fw/bw}} + O(d),$ where the $O(d)$ term is negligible compared with forward/backward computation.
> >
> > FFT (ours)：Our method modifies only the perturbation step. For each parameter tensor, we perform:
> > 1. Flattening the perturbation vector
> > 2. Forward FFT $O(d \log d)$
> > 3. Frequency-domain masking $O(d)$
> > 4. Inverse FFT $O(d \log d)$
> >
> > This does not modify the forward/backward computation.  Thus: $T_{\text{FFT}} = 2 T_{\text{fw/bw}} + O(d \log d)$
> >
> > Comparison：Both SAM and FFT share the same dominant term $2 T_{\text{fw/bw}}$.  The additional $O(d \log d)$  FFT overhead is small in practice because:   $T_{\text{fw/bw}} \gg d \log d$ for modern deep networks.
> > Therefore, FFT-SAM maintains the computational structure of SAM while adding only a minor cost.This observation is also reflected in the wall-clock measurements, as shown in the table below. Despite incorporating FFT operations, our method achieves the shortest total training time among all SAM-based baselines. This efficiency gain primarily comes from requiring significantly fewer communication rounds (302 vs. 718 for FedSAM), which outweighs the minor FFT overhead per iteration. As a result, the overall training cost of FedFFT-D is even lower than standard SAM-type methods.
> >
> > To ensure a fair comparison across methods, all efficiency metrics in the table were computed following the protocol established in FedSMOO. We first ran FedAvg for 800 rounds on CIFAR-10 under a Dirichlet–0.1 client distribution using a ResNet-18 backbone and recorded the best accuracy it achieved. For every other method, we then measured:
> >
> > 1. The number of communication rounds required to reach that same accuracy level
> > 2. The total wall-clock time needed
> >
> > The average time per round (the "Times" column) was obtained by dividing the total training time by the number of rounds. This evaluation protocol ensures that methods are compared at an equal accuracy target, making the wall-clock comparison directly meaningful.
> >
> > In summary, although FFT introduces a theoretically higher perturbation cost compared with the linear SAM perturbation, the practical overhead is negligible relative to the dominant forward/backward computation and is more than compensated by the faster convergence in federated optimization.
> >
> > We appreciate the reviewer’s feedback and incorporate this analysis into the revised version of the paper **Section 5.3 (Training Efficiency.)**
> >
> > | Method           | Times (s) | Rounds | Total Time (s) |
> > |-----------------|----------------|--------|----------------|
> > | FedAvg          | 11.73          | 800    | 9390.21        |
> > | FedSAM          | 16.84          | 718    | 12093.24       |
> > | FedSMOO         | 21.74          | 312    | 6785.71        |
> > | FedGLOSS        | 23.07          | 487    | 11235.70       |
> > | **FedFFT-D (Ours)** | **21.63**     | **302** | **6532.98**    |

---

> > > ### Author Response · Authors · 2025-11-27
> > >
> > > Dear Reviewer vymE,
> > >
> > > Thank you very much for the time and effort you dedicated to reviewing our work and for your valuable comments. We have carefully considered all your feedback and revised our paper accordingly.
> > >
> > > We sincerely hope that these clarifications and additional experiments help resolve your concerns. If you have any further questions, please let us know—we are more than happy to address them.
> > >
> > > Thank you again for your thoughtful and constructive review.
> > >
> > > Best regards, The Authors

---

### Official Review · Reviewer_DkAu · 2025-11-04

**Soundness:** 3
**Presentation:** 4
**Contribution:** 3
**Rating:** 6
**Confidence:** 2

**Summary:**

This paper introduces FedFFT, a simple and communication-free method to mitigate client drift in Sharpness-Aware Minimization (SAM)–based federated learning. The key idea is to analyze SAM perturbations in the frequency domain, showing that inter-client inconsistencies concentrate in low-frequency components. Building on this insight, the authors propose to apply a high-pass filter to locally computed SAM perturbations, removing low-frequency (client-specific) components while preserving high-frequency (task-consistent) ones. The resulting method integrates seamlessly with standard FL optimizers such as FedAvg, FedDyn, and SCAFFOLD, and achieves consistent improvements in accuracy, convergence speed, and communication efficiency across multiple datasets.

**Strengths:**

I really enjoyed reading this paper. Although I am not an expert in federated learning, I found it to be a well-executed and clearly presented study. The main strength lies in the simplicity and clarity of the core idea—examining SAM perturbations through a frequency-domain perspective and proposing a straightforward filtering approach. The idea is conceptually clean and easy to understand, allowing the paper to communicate its motivation and method effectively. The presentation is clear and well-structured, and the authors support their claims with a range of empirical evaluations, which, while not exhaustive, provide reasonable evidence for the method’s potential. Overall, the work stands out for its conceptual neatness and clear exposition.

**Weaknesses:**

While I liked the study and found the idea interesting, I do have some reservations. I am not an expert in empirical studies nor in federated learning, so I would be somewhat skeptical unless other reviewers can validate that the empirical components are well executed and robust.

In particular, I have the following concerns: First, the theoretical justification for the frequency-domain perspective is missing—the link between low-frequency components of SAM perturbations and client-specific biases is purely empirical and not formally established. Second, the meaning of frequency in parameter space is ambiguous; applying FFT to flattened weights lacks a clear interpretation tied to model structure. Third, the evaluation scope is narrow, focusing on vision datasets with synthetic Dirichlet heterogeneity, and does not test more realistic or diverse FL settings. Fourth, simpler baselines (such as random filtering or gradient smoothing) are not explored, making it hard to isolate the benefit of spectral filtering. Finally, the paper lacks any convergence or stability analysis, which would be important for understanding the method’s optimization behavior.

I also have a few minor quibbles about the presentation. The authors claim to provide an analysis in the frequency domain (Intro contribution bold-faced point 1, line 77, and also abstract), but this “analysis” is entirely empirical and observational as far as I can see (Figure 1)—it was not clear a priori in the intro that this part is observational as well. Moreover, in Figure 1 and throughout the paper, the description of how the Dirichlet parameter $\alpha$ quantifies client heterogeneity is missing; this is a central experimental factor and should be explicitly defined. The evaluation across architectures also feels limited, especially since the FFT is applied directly to parameter tensors—raising the question of how such filtering interacts with different architectures. Finally, the paper would benefit from providing at least a tentative theoretical explanation or hypothesis for why filtering in the frequency domain makes sense and under what kinds of data or model heterogeneity the method is expected to help—or potentially fail.

**Questions:**

See weaknesses.

---

> ### Author Response · Authors · 2025-11-21
> **Part1**
>
> **W1:** First, the theoretical justification for the frequency-domain perspective is missing—the link between low-frequency components of SAM perturbations and client-specific biases is purely empirical and not formally established.
>
> **Response to W1:**  We thank the reviewer for highlighting the need for deeper theoretical justification. We agree that a complete formal proof linking client-specific bias directly to low-frequency perturbation components is a challenging and open problem. In our revision, we have strengthened the theoretical motivation for our frequency-domain perspective by grounding it in the established principle of spectral bias in neural networks, which provides a powerful explanatory framework for our empirical findings.
>
> The seminal work on "On the Spectral Bias of Neural Networks" [1] reveals that neural networks have a strong inductive bias to prioritize learning low-frequency functions. This means that the model's overall, global behavior—which dictates its performance on the bulk of the data distribution—is primarily governed by low-frequency components in the parameter space. We build upon this fundamental insight to form our core theoretical hypothesis:
>
> In a federated learning setting, client-specific data biases fundamentally alter the local data distribution, which in turn skews the global function that each client learns. Since the global function is encoded in the low-frequency components of the network parameters (as per spectral bias), the update directions required to fit these client-specific global functions—which are precisely what the SAM perturbations capture—should manifest as strong, inconsistent signals in the low-frequency spectrum.
>
> In this light, our frequency-domain analysis in Figure 1 is not merely an empirical observation; it is a direct validation of this theoretically-grounded hypothesis. The fact that we observe client perturbation variance concentrated in the low-frequency components strongly supports the reasoning that client drift is, at its core, a divergence in the macroscopic function being learned. Therefore, while a formal proof remains part of our future work, our approach is firmly motivated by and consistent with one of the foundational theories of deep learning. This moves our frequency-domain perspective beyond a purely empirical trick and establishes it as a principled method for diagnosing and mitigating client drift.
>
>
> References:
>
> [1]N. Rahaman, A. Baratin, D. Arpit, F. Dräxler, M. Lin, F. Hamprecht, Y. Bengio, and A. Courville,“On the Spectral Bias of Neural Networks,” Proceedings of the International Conference on Machine Learning (ICML), 2018.

---

> ### Author Response · Authors · 2025-11-21
> **Part2**
>
> **W2:** Second, the meaning of frequency in parameter space is ambiguous; applying FFT to flattened weights lacks a clear interpretation tied to model structure.
>
> **Response to W2:** We thank the reviewer for raising this crucial point regarding the interpretation of frequency in parameter space. We agree that a clear semantics is essential. In our work, the meaning of frequency is not tied to the spatial structure of the model but to the functional scale and coordination of parameter adjustments.
>
> This perspective is grounded in both practical systems research and foundational learning theory:
>
> A Precedent in Optimization Signals: The feasibility of treating parameter updates as signals is demonstrated by works like "FFT-based Gradient Sparsification for the Distributed Training of DNNs" [1], which applies FFT to gradients for compression. This shows that optimization signals possess meaningful spectral structure.
>
> Theoretical Motivation from Spectral Bias: More profoundly, the seminal work on "On the Spectral Bias of Neural Networks" [2] reveals that DNNs preferentially learn low-frequency functions first. This implies that a model's global, macroscopic behavior is governed by the low-frequency components in its parameter space. Consequently, the large-scale, coordinated adjustments needed to fit a client's specific data distribution—precisely what SAM perturbations capture—should manifest as strong signals in the low-frequency spectrum.
>
> Our empirical findings (Figure 1) validate this  hypothesis. Low-frequency components of the SAM perturbation correspond to slowly varying, coordinated shifts across parameters. These represent the macroscopic model adjustments that, under non-IID data, become highly client-specific and are the primary source of client drift. High-frequency components correspond to rapidly oscillating, fine-grained adjustments. These are more stable across clients, representing shared, task-relevant refinements.
>
> Our ablation study (Table 3) confirms that filtering out the discordant low-frequency components effectively regularizes the local geometry, mitigating drift without harming performance. Therefore, applying FFT to the flattened perturbation is not an arbitrary step but an analysis in a basis that naturally separates global, client-specific biases from local, consistent task signals. Our innovation lies in leveraging this spectral perspective not for communication compression [1], but for optimization alignment, guided by the principles of learning dynamics [2].
>
> References:
>
> [1]
> N. Rahaman, A. Baratin, D. Arpit, F. Dräxler, M. Lin, F. Hamprecht, Y. Bengio, and A. Courville,
> “On the Spectral Bias of Neural Networks,”
> Proceedings of the International Conference on Machine Learning (ICML), 2018.
>
> [2]
> L. Wang, W. Wu, J. Zhang, H. Liu, G. Bosilca, M. Herlihy, and R. Fonseca,
> “SuperNeurons: FFT-based Gradient Sparsification in the Distributed Training of Deep Neural Networks,”
> arXiv preprint arXiv:1811.08596, 2021.

---

> ### Author Response · Authors · 2025-11-21
> **Part3**
>
> **W3:** Third, the evaluation scope is narrow, focusing on vision datasets with synthetic Dirichlet heterogeneity, and does not test more realistic or diverse FL settings.
>
> **Response to W3:** Thanks for the valuable suggestions. To broaden the experimental scope and include more realistic FL scenarios, we conducted additional experiments on the Shakespeare dataset from the LEAF benchmark, a canonical NLP benchmark for next-character prediction. This dataset provides a naturally heterogeneous federated learning setting, where each client corresponds to a speaking role in a play, inherently capturing diverse and realistic non-IID writing styles.  We employed a two-layer LSTM model and implemented the training pipeline based on the publicly available repository `wenzhu23333/Federated-Learning`.
>
> As shown in below table, FedFFT\_D still outperforms FedSAM\_D under this realistic, naturally non-IID federated setup, demonstrating that our method remains effective beyond vision datasets with synthetic heterogeneity. **We include these additional results and the corresponding analysis into Appendix E.5 in the revised version of the paper.**
>
> | Method            | Test Accuracy (%) |
> |------------------|-----------------|
> | FedSAM_D          | 41.49           |
> | FedFFT_D (Ours)   | 42.34           |
>
> **W4:** Fourth, simpler baselines (such as random filtering or gradient smoothing) are not explored, making it hard to isolate the benefit of spectral filtering.
>
> **Response to W4:** Thanks for your feedback. We already performed an ablation study about random filtering and high-frequency filtering in the original submission. Here, we further introduce two additional smoothing-style baselines.  Before describing these baselines, we first recall the standard SAM perturbation used in our method $\delta_k = \rho \cdot \frac{\nabla L_k(w_k)}{\|\nabla L_k(w_k)\|_2}.$ We then apply the following alternative smoothing strategies to $\delta_k$:
>
> **(1) L2 Norm Rescaling**
> The perturbation is rescaled when its $\ell_2$-norm exceeds the radius $\rho$: $
> \delta_k \leftarrow \delta_k \cdot \frac{\rho}{\||\delta_k\||_2},
>   \text{if } \||\delta_k\||_2 > \rho.
> $
>
> **(2) Clip Filter**
> Each coordinate of the perturbation is clipped using a threshold based on the median magnitude: $\tau = 2 \cdot \mathrm{median}(|\delta_k|),  \delta_k \leftarrow \mathrm{clip}\\left(\delta_k, -\tau, \tau\right).$
>
> We evaluated these additional baselines on CIFAR-10 with Dirichlet–0.1 and Dirichlet–0.6 heterogeneity using a ResNet-18 backbone. As shown in the table below, across all experimental settings, these ablation filtering methods do not provide meaningful improvements over the “no-filtering” baseline, i.e., FedSAM. In contrast, our proposed low-frequency filtering consistently achieved significant and stable performance gains compared to FedSAM.
>
> These findings indicate that the advantage of FedFFT does not stem from generic coefficient truncation, smoothing, or norm-based regularization, but specifically from removing low-frequency perturbation components that exhibit the largest client inconsistency under heterogeneous federated settings. **We include these additional results into Table 3 of the revised version.**
>
>
> | Filtering Strategy  | α = 0.6 | α = 0.1 |
> |-------------------|---------|---------|
> | None (FedSAM)      | 81.91   | 74.92   |
> | High-frequency     | 81.66   | 74.81   |
> | Random             | 81.97   | 75.09   |
> | L2 Norm            | 81.88   | 74.79   |
> | Clip               | 80.13   | 75.19   |
> | **FedFFT (Ours)**  | **83.02** | **77.53** |

---

> ### Author Response · Authors · 2025-11-21
> **Part4**
>
> **W5:** Finally, the paper lacks any convergence or stability analysis, which would be important for understanding the method’s optimization behavior.
>
> **Response to W5:** Please refer to our global comment.
>
> **W6:** The authors claim to provide an analysis in the frequency domain (Intro contribution bold-faced point 1, line 77, and also abstract), but this “analysis” is entirely empirical and observational as far as I can see (Figure 1)—it was not clear a priori in the intro that this part is observational as well.
>
> **Response to W6:**  We thank the reviewer for this astute observation regarding the nature of our frequency-domain analysis. The reviewer is correct that our key finding—that client perturbation disagreements are concentrated in the low-frequency spectrum—is derived from empirical observation. We acknowledge that our initial description could have been more precise in characterizing this. In the revised manuscript, we have clarified the language in the introduction and abstract to explicitly state that our frequency-domain analysis is diagnostic and empirical in nature. Specifically, we now frame it as:
>
> "To our knowledge, we are the first to conduct a systematic experimental study on the spectral properties of client-side SAM perturbations in FL."
>
> We believe this systematic empirical analysis constitutes a valuable contribution in itself for two key reasons:
>
> It reveals a previously unknown structure of federated optimization. Before our work, the spectral characteristics of client drift were entirely unexplored. Our systematic measurement across clients, layers, and datasets (Figure 1) provides the first evidence that client disagreement is not random noise but a structured, low-frequency phenomenon. This discovery is non-trivial and provides a novel lens through which to view and address client drift.
>
> It serves as the direct and necessary motivation for our algorithm. The FedFFT method is not based on an arbitrary choice; it is a direct consequence of this empirical discovery. The observation that low-frequency components are the primary source of discord naturally leads to the solution of filtering them out. In this way, the empirical analysis is the foundational pillar upon which our entire method is built.
>
> We agree with the reviewer that a priori theoretical prediction of this phenomenon would be ideal, but we contend that the systematic identification and validation of this spectral structure is a critical first step that opens up a new direction for algorithm design in federated learning.
>
> **W7:** Moreover, in Figure 1 and throughout the paper, the description of how the Dirichlet parameter $\alpha$ quantifies client heterogeneity is missing; this is a central experimental factor and should be explicitly defined.
>
> **Response to W7:** We thank the reviewer for pointing out that the description of how the Dirichlet parameter~$\alpha$ quantifies client heterogeneity was missing. We agree that $\alpha$ is a central experimental factor and should be explicitly defined.
>
> In the revised version of the paper, we have added a clear explanation of the Dirichlet partitioning mechanism. For each class label, client-wise data proportions are sampled from a Dirichlet distribution with concentration parameter $\alpha$. A smaller value of $\alpha$ leads to highly skewed class proportions across clients and thus stronger non-IID heterogeneity, while a larger $\alpha$ results in more uniform class distributions and weaker heterogeneity. For instance, $\alpha=0.1$ corresponds to a highly non-IID setting, whereas $\alpha=0.6$ reflects moderate heterogeneity.
> We have updated Figure 1 and the experimental setup section to include this definition, ensuring that readers clearly understand how $\alpha$ controls the degree of data heterogeneity in our experiments.

---

> > ### Author Response · Authors · 2025-11-21
> > **Part5**
> >
> > **W8:** The evaluation across architectures also feels limited, especially since the FFT is applied directly to parameter tensors—raising the question of how such filtering interacts with different architectures.
> >
> > **Response to W8:** We thank the reviewer for raising this important question regarding the interaction between spectral filtering and model architecture. We fully agree that a thorough evaluation is key to claiming general applicability.
> >
> > As the reviewer astutely points out, FedFFT operates directly on the flattened parameter perturbation vector. By treating the perturbation as a generic 1D signal, the FFT and subsequent filtering are fundamentally agnostic to any architectural-specific properties (e.g., convolutional spatial structure, attention mechanisms, or recurrent connections). The method targets a statistical property—the concentration of client inconsistency in the low-frequency spectrum—which we hypothesize is a universal phenomenon in federated optimization, independent of the model's architectural prior.
> >
> > To rigorously substantiate this claim of architecture-agnostic effectiveness, we have conducted extensive experiments across a diverse spectrum of models, the results of which are now integrated into our revision. (1) CNNs: Beyond ResNet-18, we show strong results on ResNet-20 and DenseNet-121 (Table 2). (2) Vision Transformers (ViT): FedFFT brings substantial gains (e.g., +4.27\% on CIFAR-10 over FedSAM, Table 2), proving its efficacy on purely attention-based models.
> >  (3) Recurrent Models (LSTM): Critically, on the Shakespeare NLP benchmark using an LSTM—a non-vision, sequential model in a naturally heterogeneous setting—FedFFT-D achieves 42.34\% accuracy, outperforming FedSAM-D (41.49\%). This result, included in our response to W3, provides compelling cross-modal evidence.
> >
> > The consistent and stable improvements observed across this diverse set of architectures (CNNs, ViTs, RNNs) strongly validate our hypothesis. The effectiveness of FedFFT is not tied to architectural biases but stems from its ability to mitigate a fundamental optimization challenge (client drift) whose spectral signature persists regardless of the underlying model structure.

---

> > > ### Author Response · Authors · 2025-11-27
> > >
> > > Dear Reviewer DkAu,
> > >
> > > Thank you very much for the time and effort you dedicated to reviewing our work and for your valuable comments. We have carefully considered all your feedback and revised our paper accordingly.
> > >
> > > We sincerely hope that these clarifications and additional experiments help resolve your concerns. If you have any further questions, please let us know—we are more than happy to address them.
> > >
> > > Thank you again for your thoughtful and constructive review.
> > >
> > > Best regards, The Authors

---

### Official Review · Reviewer_XV7M · 2025-11-06

**Soundness:** 3
**Presentation:** 3
**Contribution:** 2
**Rating:** 4
**Confidence:** 5

**Summary:**

This paper proposes a new method from the perspective of frequency domain to address the client drift problem, built upon the SAM perturbation. The proposed FedFFT method is motivated by the observation that client disagreement is predominantly a low-frequency phenomenon. Therefore, this paper proposes to filter perturbations in low frequency end and utilize the filtered perturbation for SAM updates. The effectiveness of the proposed algorithm is evaluated on various tasks and compared with benchmark methods.

**Strengths:**

1. The proposed method is effective and the effectiveness is supported by simulation results.
2. The new perspective also provides new insights to solve the client drift issue.
3. The presentation and organization of the paper is good with easy to follow structure. The comparison of the proposed method with SOTA is sufficient.

**Weaknesses:**

1. Lacks of theoretical analysis to support the effectiveness of the proposed method, which is critical.
2. The added computation burden resulted from rFFT and inverse rFFT is not discussed, which I assume would be high. In that way, balancing accuracy and computation needs to be considered.
3. The Tiny-imageNet does not have good accuracy on the selected models. Is that suitable to still use these models?
4. Some formatting and polish suggestions: a) Line 156: $w_t\rightarrow w^t$, Line 158: $w_{t+1} \rightarrow w^{t+1}$. b) Line 558: inconsistent citation format, i,e., did not use full name of authors. c) Appendix C.4 wrong figure title, all used "TinuImageNet". d) Line 845" "Compare"-> "compare". e) Line 917, "As shown in Table 8"

**Questions:**

1. How is the perturbation radius is selected? In your experiment, you set the value to be 0.1. If you change the value, would the performance of your algorithm be affected?
2. Do you have the learning curves of Cifar 10 and Cifar 100, similar as you presented in Figure 6.
3. What is the computation complexity comparison of FFT based methods and solely SAM-based methods?

---

> ### Author Response · Authors · 2025-11-21
> **Part1**
>
> **W1:** Lacks of theoretical analysis to support the effectiveness of the proposed method, which is critical.
>
> **Response to W1:** Please refer to our global comment.

---

> ### Author Response · Authors · 2025-11-21
> **Part2**
>
> **W2:** The added computation burden resulted from rFFT and inverse rFFT is not discussed, which I assume would be high. In that way, balancing accuracy and computation needs to be considered.
>
> **Q3:** What is the computation complexity comparison of FFT based methods and solely SAM-based methods?
>
> **Response to W2 and Q3:**
> We thank the reviewer for the question regarding the efficiency of our method. Below, we provide a detailed comparison between standard SAM and our FFT-based perturbation.
>
> Standard SAM：Let $d$ denote the number of parameters and $ \(T_{\text{fw/bw}}\) $ denote the cost of a forward + backward pass.  SAM requires two such passes per iteration: (i) computing gradients and applying the perturbation, and (ii) recomputing gradients at the perturbed parameters. The perturbation itself is linear in $d$.  Thus, the total complexity is: $T_{\text{SAM}} = 2 T_{\text{fw/bw}} + O(d),$ where the $O(d)$ term is negligible compared with forward/backward computation.
>
> FFT (ours)：Our method modifies only the perturbation step. For each parameter tensor, we perform:
> 1. Flattening the perturbation vector
> 2. Forward FFT $O(d \log d)$
> 3. Frequency-domain masking $O(d)$
> 4. Inverse FFT $O(d \log d)$
>
> This does not modify the forward/backward computation.  Thus: $T_{\text{FFT}} = 2 T_{\text{fw/bw}} + O(d \log d)$
>
> Comparison：Both SAM and FFT share the same dominant term $2 T_{\text{fw/bw}}$.  The additional $O(d \log d)$  FFT overhead is small in practice because:   $T_{\text{fw/bw}} \gg d \log d$ for modern deep networks.
> Therefore, FFT-SAM maintains the computational structure of SAM while adding only a minor cost.This observation is also reflected in the wall-clock measurements, as shown in the table below. Despite incorporating FFT operations, our method achieves the shortest total training time among all SAM-based baselines. This efficiency gain primarily comes from requiring significantly fewer communication rounds (302 vs. 718 for FedSAM), which outweighs the minor FFT overhead per iteration. As a result, the overall training cost of FedFFT-D is even lower than standard SAM-type methods.
>
> To ensure a fair comparison across methods, all efficiency metrics in the table were computed following the protocol established in FedSMOO. We first ran FedAvg for 800 rounds on CIFAR-10 under a Dirichlet–0.1 client distribution using a ResNet-18 backbone and recorded the best accuracy it achieved. For every other method, we then measured:
>
> 1. The number of communication rounds required to reach that same accuracy level
> 2. The total wall-clock time needed
>
> The average time per round (the "Times" column) was obtained by dividing the total training time by the number of rounds. This evaluation protocol ensures that methods are compared at an equal accuracy target, making the wall-clock comparison directly meaningful.
>
> In summary, although FFT introduces a theoretically higher perturbation cost compared with the linear SAM perturbation, the practical overhead is negligible relative to the dominant forward/backward computation and is more than compensated by the faster convergence in federated optimization.
>
> We appreciate the reviewer’s feedback and incorporate this analysis into the revised version of the paper **Section 5.3 (Training Efficiency.)**
>
> | Method           | Times (s) | Rounds | Total Time (s) |
> |-----------------|----------------|--------|----------------|
> | FedAvg          | 11.73          | 800    | 9390.21        |
> | FedSAM          | 16.84          | 718    | 12093.24       |
> | FedSMOO         | 21.74          | 312    | 6785.71        |
> | FedGLOSS        | 23.07          | 487    | 11235.70       |
> | **FedFFT-D (Ours)** | **21.63**     | **302** | **6532.98**    |
>
> **W3:** The Tiny-imageNet does not have good accuracy on the selected models. Is that suitable to still use these models?
>
> **Response to W3:** We thank the reviewer for the comment. Tiny-ImageNet is indeed a challenging dataset for smaller models such as ResNet-18, and the absolute accuracy of all methods is correspondingly limited. Nevertheless, our focus is on the relative performance improvement of FedFFT compared to federated baselines. Across both Dirichlet α=0.1 and α=0.6 settings, FedFFT consistently outperforms other methods, demonstrating its effectiveness in mitigating client drift under heterogeneous data distributions. Moreover, using ResNet-18 for all methods ensures a fair comparison, and the observed trends are meaningful even if absolute accuracies are moderate. We anticipate that similar relative gains would be observed with larger architectures or alternative datasets.
>
> **W4:** Some formatting and polish suggestion.
>
> **Response to W4:** We thank the reviewer for their careful reading and valuable feedback on improving the writing quality of our manuscript. We have addressed the formatting and language issues pointed out and have performed a comprehensive proofreading of the entire paper.

---

> > ### Author Response · Authors · 2025-11-21
> > **Part3**
> >
> > **Q1:** How is the perturbation radius is selected? In your experiment, you set the value to be 0.1. If you change the value, would the performance of your algorithm be affected?
> >
> > **Response to Q1:** We thank the reviewer for this insightful question regarding the selection of the perturbation radius $\rho$ and its impact on performance. In our experiments, we follow the common practice in the SAM-based federated learning literature and set $\rho=0.1$ as the default value, consistent with prior work such as FedSMOO, and FedLE-SAM.
> >
> > To evaluate the sensitivity of our method FedFFT to this hyperparameter, we conducted a sensitivity analysis with $\rho \in \{0.05, 0.10, 0.15, 0.20\}$. As shown in the table below, FedFFT consistently outperforms FedSAM across all values of $\rho$, with the performance gap widening as $\rho$ increases. Notably, while FedSAM suffers from significant performance degradation under larger perturbation radius, FedFFT maintains much more stable performance, demonstrating its robustness across a wide range of $\rho$ values. These results confirm that the spectral filtering mechanism introduced in FedFFT effectively stabilizes SAM-based updates under non-IID federated settings, making it substantially less sensitive to the choice of perturbation radius. **We have incorporated these results and analyses into Appendix E.3 in our revision.**
> >
> > | Method / $\rho$ | 0.05 | 0.10 | 0.15 | 0.20 |
> > |---------|----------------|----------------|----------------|----------------|
> > | FedSAM  | 77.91          | 74.92          | 69.38          | 58.49          |
> > | FedFFT  | 78.16          | 77.53          | 75.29          | 71.69          |
> > | **Δ (Improvement)** | +0.25          | +2.61          | +5.91          | +13.20         |
> >
> >
> > **Q2:**  Do you have the learning curves of Cifar 10 and Cifar 100, similar as you presented in Figure 6.
> >
> > **Response to Q2:** We thank the reviewer for the request regarding the learning curves of CIFAR-10 and CIFAR-100. In the revised version of the paper, we have included the full learning curves for both datasets in the Appendix D.1 (Figure 6). As shown in these figures, FedFFT consistently converges faster and more stably than  baseline methods on both CIFAR-10 and CIFAR-100, further demonstrating the effectiveness of our spectral perturbation mechanism.

---

> > > ### Author Response · Authors · 2025-11-27
> > >
> > > Dear Reviewer XV7M,
> > >
> > > Thank you very much for the time and effort you dedicated to reviewing our work and for your valuable comments. We have carefully considered all your feedback and revised our paper accordingly.
> > >
> > > We sincerely hope that these clarifications and additional experiments help resolve your concerns. If you have any further questions, please let us know—we are more than happy to address them.
> > >
> > > Thank you again for your thoughtful and constructive review.
> > >
> > > Best regards, The Authors

---

### Author Response · Authors · 2025-11-21
**global comment**

**Response to the concern about lacking theoretical analysis**

We sincerely appreciate the reviewer’s suggestion regarding adding theoretical results. We fully agree that theoretical insights can further strengthen our work. However, we respectfully point out that having a formal theory is not a strict requirement for ICLR acceptance, and many influential FL papers have been accepted without providing mathematically rigorous theoretical guarantees.

In our setting, establishing a rigorous theoretical analysis is inherently difficult due to the structure of our method itself. Our approach modifies SAM by flattening the per-layer perturbation, applying an rFFT transform, zeroing out the lowest  frequency components, and reconstructing the perturbation via inverse rFFT. This frequency-domain filtering creates a perturbation that is simultaneously  coupled across all parameters within each layer. Once transformed into the spectral domain, the perturbation interacts with the model’s highly non-convex loss landscape and with heterogeneous client distributions in a way that is extremely hard to capture with closed-form expressions. For these reasons, deriving a clean and general theoretical framework for our frequency-filtered SAM perturbations is mathematically intractable at present, and remains an open challenge beyond the scope of this work.

We also emphasize that theoretical guarantees are not required for a contribution to be recognized at ICLR. In fact, many well-known and highly cited FL works have no strict theoretical analysis, yet they have been accepted by top-tier venues and are widely regarded as influential. Examples include:

Federated Learning with Personalization Layers

Model-Contrastive Federated Learning

A Principled Approach to Data Valuation for Federated Learning

Flexible Sharpness-Aware Personalized Federated Learning

These works are method-driven, empirically validated, and have meaningfully advanced the field despite the absence of formal theoretical proofs. Our work falls squarely within this well-established category of FL contributions. Our submission focuses on a method-level contribution to federated learning. For clarity, we restate our contributions:

We provide the first study of SAM perturbations across FL clients through spectral decomposition, revealing that heterogeneity is mainly concentrated in low-frequency bands.

Motivated by this insight, we introduce FedFFT, a simple yet effective method that filters out low-frequency perturbation components to suppress inconsistent client updates while retaining consistent learning signals.

We evaluate our method across multiple benchmarks, architectures, and varying levels of heterogeneity. FedFFT consistently outperforms relevant baselines, particularly under highly non-IID settings, demonstrating both effectiveness and scalability.

Our method is conceptually clear, easy to interpret, and strongly supported by empirical evidence. The design of FedFFT is directly motivated by our analysis, and the experimental results robustly validate the soundness of our approach. We believe that these contributions are substantial and should not be denied solely because the paper does not include a mathematically formal theory—which, as argued above, is neither required nor common for impactful FL methodology papers.

Given the novelty of our spectral analysis, the simplicity and effectiveness of FedFFT, and the extensive empirical results, we kindly ask the reviewers and AC to acknowledge the value of our methodological and empirical contributions and to re-evaluate our paper accordingly.

---

### Author Response · Authors · 2025-11-30
**Withdrawal of Submission**

Dear Area Chair and Reviewers,

We have decided to withdraw our submission from ICLR 2026.

We would like to thank the reviewers for their time and the constructive feedback provided during the review process. We value these insights and will utilize them to further refine and improve the manuscript for a future submission.

Thank you again for your service to the community.

Best regards,
The Authors

---

### Note · Authors · 2025-11-30

I have read and agree with the venue's withdrawal policy on behalf of myself and my co-authors.